# A fungal extracellular effector inactivates plant polygalacturonase-inhibiting protein

Wei Wei[1,8], Liangsheng Xu [2,8], Hao Peng [3], Wenjun Zhu[4], Kiwamu Tanaka [1,5], Jiasen Cheng[6], Karen A. Sanguinet[3,5], George Vandemark[1,7] & Weidong Chen [1,3,5,7 ✉]

Plant pathogens degrade cell wall through secreted polygalacturonases (PGs) during infection. Plants counteract the PGs by producing PG-inhibiting proteins (PGIPs) for protection, reversibly binding fungal PGs, and mitigating their hydrolytic activities. To date, how fungal pathogens specifically overcome PGIP inhibition is unknown. Here, we report an effector, *Sclerotinia sclerotiorum* PGIP-INactivating Effector 1 (SsPINE1), which directly interacts with and functionally inactivates PGIP. *S. sclerotiorum* is a necrotrophic fungus that causes stem rot diseases on more than 600 plant species with tissue maceration being the most prominent symptom. SsPINE1 enhances *S. sclerotiorum* necrotrophic virulence by specifically interacting with host PGIPs to negate their polygalacturonase-inhibiting function via enhanced dissociation of PGIPs from PGs. Targeted deletion of *SsPINE1* reduces the fungal virulence. Ectopic expression of *SsPINE1* in plant reduces its resistance against *S. sclerotiorum*. Functional and genomic analyses reveal a conserved virulence mechanism of cognate PINE1 proteins in broad host range necrotrophic fungal pathogens.

[1] Department of Plant Pathology, Washington State University, Pullman, WA 99164, USA. [2] State Key Laboratory of Crop Stress Biology for Arid Areas, College of Plant Protection, Northwestern A&F University, Yangling 712100 Shaanxi, China. [3] Department of Crop & Soil Sciences, Washington State University, Pullman, WA 99164, USA. [4] School of Life Science and Technology, Wuhan Polytechnic University, Wuhan, Hubei 430023, China. [5] Molecular Plant Science Program, Washington State University, Pullman, WA 99164, USA. [6] State Key Laboratory of Agricultural Microbiology, Huazhong Agricultural University, Wuhan, China. [7] USDA Agricultural Research Service, Grain Legume Genetics and Physiology Research Unit, Pullman, WA 99164, USA. [8] These authors contributed equally: Wei Wei, Liangsheng Xu. ✉email: weidong.chen@usda.gov

There is a continuous arms race between fungal pathogens and their plant hosts that is best illustrated by the classical zig-zag model[1]. An intense battle of attack, counterattack and counter-counterattack occurs at the apoplastic space that often determines the outcome of plant-pathogen interactions[2–4]. Enzyme-inhibitor interactions often dominate in the apoplast battleground[5,6]. Since plant cell wall is a major barrier to intrusion by pathogenic microorganisms, pathogens secret an array of cell wall-degrading enzymes (CWDEs) to compromise cell wall integrity in order to gain access[7]. An important part of the CWDEs is pectin-degrading polygalacturonases (PGs) as pectin is an important component of cell wall and the middle lamella. Necrotrophic pathogens often possess multiple PGs in their genomes[8]. For example, the polyphagous necrotrophic pathogen *S. sclerotiorum* has a genome that encodes at least five endo-polygalacturonases that are expressed at different infection stages and conditions. PGs play important roles in virulence as demonstrated using gene-deletion in *Botrytis cinerea*[9] and by enzymatic and expression studies in *S. sclerotiorum*[10,11]. In defense, all plants studied thus far have evolved to have PGIPs anchored to the cell wall that are capable of binding to fungal PGs and mitigating their hydrolytic activities, providing the first line of defense against fungal infection[12,13].

PGIPs are a cell wall-bound, leucine-rich repeat super family of proteins[12,13], a conserved structure feature of plant disease resistance proteins and for protein-protein interactions[14]. PGIPs were first discovered in 1971 from cell walls of bean, tomato and cell culture of sycamore trees[15]. They have since been shown to occur in all plants and are often encoded by multi gene families[13]. PGIPs function in limiting fungal invasion in two ways: as inhibitors and regulators of PG activity and by allowing accumulation of short-chained oligogalacturonides (products of partial degradation of plant pectin by the fungal PGs) that are recognized as damage-associated molecular patterns in eliciting plant defense response[12,16–18]. Over-expressing PGIPs often reduces disease severity, demonstrating the role of PGIPs in enhancing plant immunity[13]. Fungal PGs are very sensitive to inhibition by PGIPs in enzyme assays. For example, *Sclerotinia sclerotiorum* SsPG6 activity was severely inhibited by *Brassica napus* BnPGIP1[19]. However, *Arabidopsis thaliana* lines over-expressing BnPGIP1 or BnPGIP2 show no observable difference in disease phenotype compared with wildtype control line, except that BnPGIP2-expressing lines showed an initial delay in disease onset[19]. Likewise, transgenic rapeseed lines overexpressing BnPGIP10 did not improve resistance to infection by *Sclerotinia sclerotiorum*[20]. These incidences of the increased expression levels of PGIPs failing to enhance resistance to *Sclerotinia* infection suggest that *S. sclerotiorum* possesses a mechanism(s) to overcome resistance provided by the PGIPs.

*S. sclerotiorum* is a necrotrophic plant pathogen with a broad host range of more than 600 plant species[21], including most dicots and some non-graminaceous monocot plants that have high pectin contents in their cell wall[14]. Indeed, *S. sclerotiorum* is responsible for significant damage of many economically important crops[22] and, despite extensive studies, the mechanisms of its necrotrophic pathogenesis are still inadequately understood. Recently Liang and Rollins[23] proposed, based on infection cytological evidence and supported by genetic investigations, a two-phase infection model in which the pathogen suppresses host basal defense reactions in the first phase and induces necrosis and macerates host tissue in the second phase. *S. sclerotiorum* secrets oxalic acid, CWDEs such as PGs and other pathogenicity factors during plant infection[22,24]. Earlier studies using UV-induced, genetically undefined mutants that concomitantly lost oxalic acid production and pathogenicity led to the conclusion that oxalic acid is the primary pathogenicity determinant[25–27]. However,

more recent studies showed that the role of oxalic acid is most likely in the second phase of infection because genetically defined oxalate-minus mutants still retained pathogenicity and remained highly virulent on many host plants[28–30]. As a non-specific toxin oxalic acid can enhance virulence by regulating autophagy and apoptotic-like programmed cell death (PCD)[27]. Because the UV-induced mutations were not determined and therefore genetically undefined, the UV-induced mutants are not ideal for elucidating virulence mechanisms[23,30]. Looking beyond oxalic acid, the *S. sclerotiorum* genome encodes dozens (>70) potentially secreted proteins which are putative pathogenicity effectors that may interact with and modulate host response to infection[31]. Some of the putative secreted effector proteins have been empirically confirmed[23,30]. For the *S. sclerotiorum* virulence associated effectors whose host targets are known, effectors SsSSVP1 and SsCP1 target host cytochrome b-c1 complex subunit 8 (QCR8) and resistant protein PR1, respectively, both of the targets are highly conserved in plants, contributing to the broad host range necrotrophy[32,33]. However, how *S. sclerotiorum* specifically overcomes PG inhibition imposed by plant PGIPs remains unknown.

Here we report a small fungal effector protein secreted by *S. sclerotiorum* that can inactivate PGIP. *Sclerotinia sclerotiorum* PGIP-inactivating effector 1 (SsPINE1) directly interacts with plant PGIPs to enhance PG-PGIP dissociation, which results in suppression of plant innate immunity and enhancement of pectin degradation, and consequently necrotrophic virulence. Deletion of *SsPINE1* reduces *S. sclerotiorum* virulence. In contrast, ectopic expression of *SsPINE1* in plant reduces plant resistance against *S. sclerotiorum*. Identification of SsPINE1 is a crucial step towards understanding the complex infection mechanism of this important fungal pathogen. Moreover, we provide evidence that the SsPINE1-mediated PGIP inactivation mechanism is evolutionarily conserved among a range of broad host range necrotrophic fungi.

## Results

**Identification of *SsPINE1*.** SsPINE1 was identified through transcriptome analysis and knockout screening of genes that are preferentially expressed at early stages of plant infection in both the wildtype *S. sclerotiorum* strain and a genetically defined oxalate-minus mutant. The oxalate-minus mutant was included in the analysis because it retained pathogenicity[28–30] despite previous claims that oxalic acid is the primary pathogenicity determinant. Candidate effector-encoding genes were selected from upregulated transcripts in both wildtype and the oxalate-minus mutant whose putative encoded proteins harbor known features of fungal effectors[34], including being shorter than 300 amino acid (aa) residues, having a signal peptide (SP), and rich in cysteine (Cys) residues. The *S. sclerotiorum* genome contains a single copy of *SsPINE1* (Ss1g_08128) (https://www.broadinstitute.org/fungal-genome-initiative/sclerotinia-sclerotiorum-genome-project). The three exons of the *SsPINE1* cDNA encode a 105 aa secreted protein with 12 Cys residues and a 21-aa SP at its N-terminus.

**Sclerotinia sclerotiorum SsPINE1 is required for full virulence.** Two *SsPINE1* knockout (Δ*SsPINE1*) mutants (*KoSsPINE1-1* and *KoSsPINE1-7*) were generated for virulence analysis (Supplementary Fig. 1a, b). Both Δ*SsPINE1* mutants showed a similar growth rate and colony morphology as those of the wild-type strain (Supplementary Fig. 1c), suggesting SsPINE1 is not required for normal growth and completion of life cycle. However, the Δ*SsPINE1* mutants exhibited significantly reduced virulence. When infecting pea plants (*Pisum sativum* var. Guido),

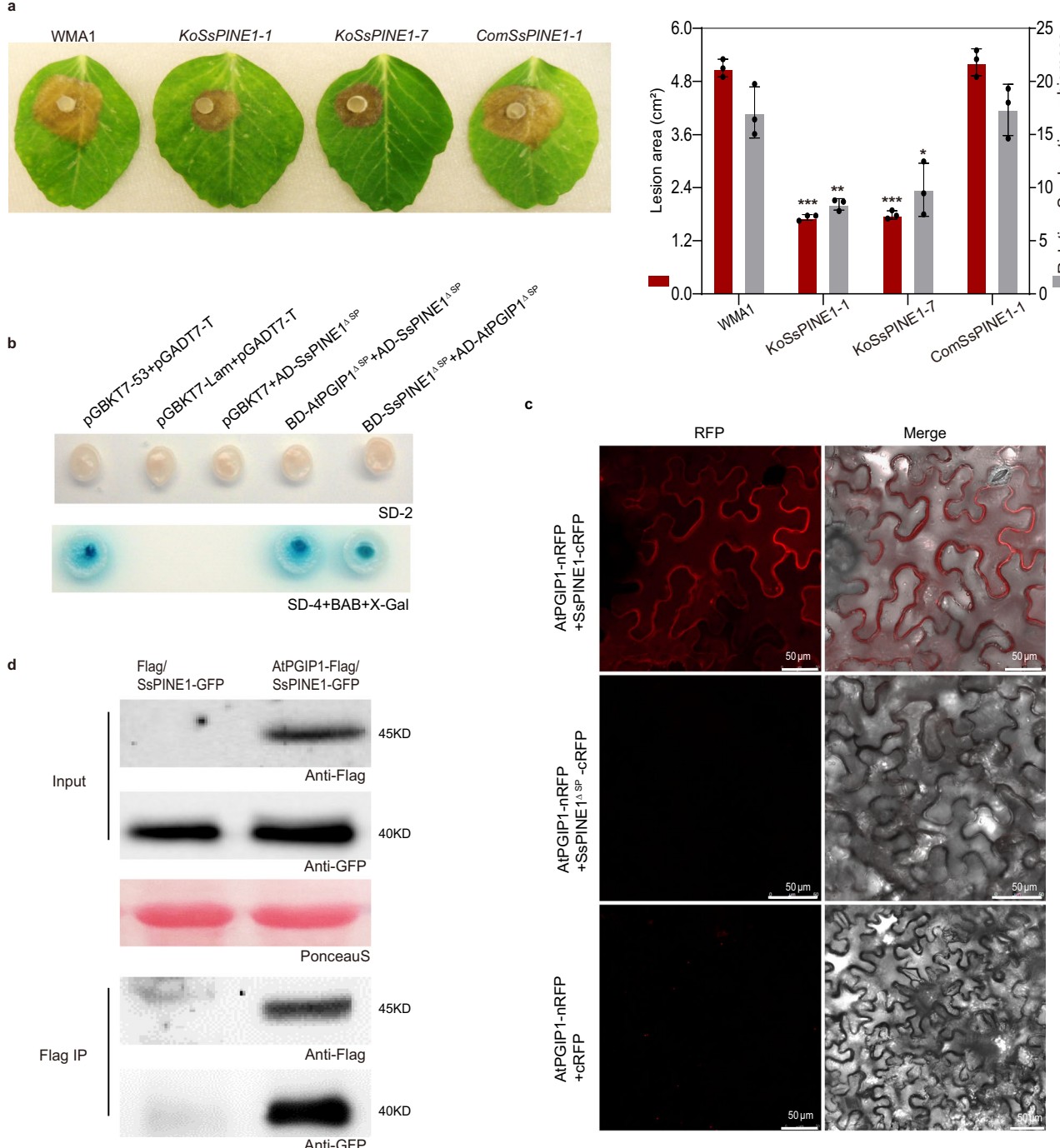

**Fig. 1 *Sclerotinia sclerotiorum* SsPINE1 is required for full virulence and interacts with AtPGIP1. a** Disease symptoms and disease quantification (lesion area and relative *S. sclerotiorum* biomass) on pea leaves caused by the wild-type strain WMA1, two *SsPINE1* mutants *KoSsPINE1-1* and *KoSsPINE1-7*, and the complement strain *ComSsPINE1-1*. Representative leaves were photographed 2 days post inoculation (dpi). Data represent mean ± s.d. of $n = 3$ biological replicates. *, ** and *** indicate significant difference from the wildtype WMA1 at $p < 0.05$, 0.01 and 0.001, respectively, in two-tailed *t*-test. The experiment was performed three times independently with similar results obtained. **b** Yeast two- hybrid assay showed SsPINE1 interacts with AtPGIP1 as either the bait or the prey. **c** Bimolecular fluorescence complementation (BiFC) assay confirmed that SsPINE1 interacts with AtPGIP1 and the interaction requires signal peptide. *N. benthamiana* leaves were agroinfiltrated with a mixture of *Agrobacterium tumefaciens* strains harboring constructs AtPGIP1-nRFP and SsPINE1-cRFP (top panel), AtPGIP1-nRFP and SsPINE1$^{\Delta SP}$-cRFP (middle panel), and the negative control AtPGIP1-nRFP and cRFP (lower panel). The RFP fluorescence was monitored at 2 days post-agroinfiltration using confocal laser scanning microscope. Representative images are shown. The experiment was performed three times with similar results obtained. **d** Co-immunoprecipitation (Co-IP) assay showed that SsPINE1 is associated with AtPGIP1 *in planta*. Co-IP constructs AtPGIP1-Flag and SePINE1-GFP were transiently expressed in *N. benthamiana* leaves. Immunoprecipitations were performed with anti-Flag agarose (Flag IP), and SsPINE1 was detected in the immuno-precipitates using anti-GFP antibody. Representative images are shown. The experiments were performed three times independently with similar results obtained. Source data are provided as a Source data file.

both ΔSsPINE1 mutants caused significantly (39 to 45%) smaller disease lesions than the wildtype strain (Fig. 1a).

In the complementation assay, the wildtype SsPINE1 allele was cloned from S. sclerotiorum cDNA, fused with GFP (SsPINE1-GFP), and introduced into the ΔSsPINE1 mutant KoSsPINE1-1. The reintroduced SsPINE1-GFP was detected in the culture filtrate of the complemented strain (Supplementary Fig. 1d), suggesting that the SsPINE1-GFP fusion protein is normally secreted. SsPINE1-GFP fully restored mutant virulence to the wild-type level (Fig. 1a), which demonstrates that SsPINE1 is a critical component for S. sclerotiorum virulence and SsPINE1-GFP is functional.

**SsPINE1 physically interacts with AtPGIP1.** Since SsPINE1 is a secreted virulence protein with structural features of known fungal effectors, it was used as a yeast two-hybrid bait to screen an Arabidopsis thaliana cDNA library for potential interacting targets in plants. A total of 130 clones were positive in two independent screens. After sequencing the 130 positive clones, 108 prey proteins were identified (Supplementary Table 1). Only five of 108 prey proteins interacted with SsPINE1 in both screens (top five in the Supplementary Table 1). Among the five interacting proteins, A. thaliana polygalacturonase-inhibitor 1 (AtPGIP1) is the only protein that has a known function[35]. Consequently, AtPGIP1 was selected for further investigation.

Targeted yeast two-hybrid assays showed the interaction between AtPGIP1 and the signal peptide-truncated SsPINE1 (SsPINE1$^{\Delta SP}$) in both combinations with SsPINE1$^{\Delta SP}$ either as a bait or as a prey (Fig. 1b). In contrast, SsPINE1$^{\Delta SP}$ did not interact AtPGIP2, the other member of the Arabidopsis thaliana PGIP two gene family[35] (Supplementary Fig. 2a). Three of the other prey proteins that showed up in both yeast 2-hybrid screens were also tested in the yeast 2-hybrid assay. SsPINE1$^{\Delta SP}$ did not interact with AtNdhL or At2G28100 but did interact with AT2G35790 (Supplementary Fig. 2b). Since AT2G35790 is a trans-membrane protein in the mitochondria with no known functions and SsPINE1 is in the apoplast, its interaction with SsPINE1 is likely an artifact of yeast 2-hybrid assay. These results suggest the high specificity of SsPINE1-AtPGIP1 interaction. In bi-molecular fluorescence complementation (BiFC) assays using red fluorescent protein (RFP), co-agroinfiltrated SsPINE1-cRFP and AtPGIP1-nRFP could interact to restore red fluorescence in Nicotiana benthamiana pavement cells. In contrast, fluorescence was not detected in negative control infiltrations or leaves co-infiltrated with SsPINE1$^{\Delta SP}$-cRFP and AtPGIP1-nRFP, which can be explained by the lack of signal peptide in SsPINE1 (Fig. 1c). As controls, fusion protein expression was detected from all BiFC constructs (Supplementary Fig. 3a).

The interaction between SsPINE1 and AtPGIP1 was further confirmed by an in vivo co-immunoprecipitation (Co-IP) assay (Fig. 1d). SsPINE1-GFP and AtPGIP1-3xFlag were co-expressed in N. benthamiana leaves by agroinfiltration. Both were co-immunoprecipitated using anti-Flag agarose (Flag IP) (Fig. 1d). All three approaches (yeast two-hybrid, BiFC and Co-IP) demonstrate that SsPINE1 physically interacts with AtPGIP1.

**SsPG1 is critical for S. sclerotiorum virulence.** Previous studies on PG-PGIP interactions suggest that S. sclerotiorum PGs are potential acting targets of AtPGIP1[35,36]. The S. sclerotiorum genome encodes at least five endo-PGs that are expressed at different infection stages or growth conditions and expression of SsPG1 precedes that of the other SsPGs[8,10,11,37]. Consistent with previous reports[11,19,37,38], our RT-qPCR analysis using S. sclerotiorum-inoculated A. thaliana showed that among four SsPGs (SsPG1, 3, 5 and 6) only SsPG1 (Ss1G_10167) was significantly

induced during early stages of infection (250-fold increase at 36 h) and had the same expression pattern as that of SsPINE1 (Supplementary Fig. 4). Therefore, SsPG1 was selected for further study.

Two SsPG1 knockout mutants (KoSsPG1-1 and KoSsPG1-3) were generated (Supplementary Fig. 5a, b). Both mutants exhibited similar growth rates and colony morphology as the wild type on PDA (Supplementary Fig. 5c, d). When poly-galacturonic acid was used as the sole carbon source, both mutants exhibited a slower growth rate when compared to the wild-type strain (Supplementary Fig. 5c), demonstrating the role of SsPG1 in securing nutrients from pectin-like substances. Both ΔSsPG1 mutants (KoSsPG1-1 and KoSsPG1-3) exhibited significantly reduced virulence on pea plants, and the reduced-virulence phenotype was restored to the level of the wildtype strain by complementation with the wild-type SsPG1 cDNA fused with GFP (SsPG1-GFP) (Fig. 2a). This gene knockout result directly demonstrates the critical role of SsPG1 in S. sclerotiorum virulence, supporting earlier conclusions based on enzymatic assays and gene expression patterns[10,11]. SsPG1-GFP was detected in cultural filtrates of the complement strain (Supplementary Fig. 5e), suggesting that SsPG1 is a secreted protein.

**SsPG1 physically interacts with AtPGIP1.** Consistent with previous studies of PG-PGIP interactions[35,36], SsPG1 physically interacts with AtPGIP1 as demonstrated by BiFC and Co-IP (Fig. 2b, c). Red fluorescence was observed in tobacco leaves co-infiltrated with AtPGIP1-nRFP and SsPG1-cRFP constructs, but not in the negative control (Fig. 2b). As controls, fusion protein expression was detected from all BiFC constructs (Supplementary Fig. 3b). In Co-IP assays with SsPG1-GFP and AtPGIP1-3xFlag as well as SsPINE1-GFP and AtPGIP1-3xFlag constructs, SsPG1-GFP was detected by Flag IP, and AtPGIP1-3xFlag was detected by GFP IP (Fig. 2c). Both the BiFC and Co-IP results demonstrate physical interaction between SsPG1 and AtPGIP1.

**SsPINE1 outcompetes SsPG1 in binding with AtPGIP1.** Since both SsPINE1 and SsPG1 interact with AtPGIP1, they may compete for binding. All three proteins were found to be co-localized in planta, as indicated by the GFP-labeled SsPG1 and the complemented red fluorescence between AtPGIP1-nRFP and SsPINE1-cRFP (Supplementary Fig. 2c).

If SsPINE1 functions in S. sclerotiorum virulence via suppressing the inhibition effect of AtPGIP1 on SsPG1, then SsPINE1 would be expected to have higher binding affinity with AtPGIP1 than SsPG1 in Co-IP assays. In a modified Co-IP experiment, three constructs SsPINE1-GFP, SsPG1-GFP and AtPGIP1-3xFlag were co-infiltrated into tobacco leaves. Flag IP detected significantly more SsPINE1-GFP than SsPG1-GFP in the co-precipitates (Fig. 3a, lane indicated by an arrow), indicating that SsPINE1 is more readily associated with AtPGIP1 than does SsPG1.

Since PG-PGIP binding is reversible[39], we tested if the addition of a competing protein could dissociate the SsPINE1/AtPGIP1 or SsPG1/AtPGIP1 complexes formed in vivo. After verification of protein expression levels (Fig. 3b), equal volumes of the same tobacco leaf lysate co-infiltrated with SsPG1-GFP and AtPGIP1-3xFlag constructs were mixed with 0, 100 ng and 10 μg of the purified SsPINE1-GFP fusion protein expressed in yeast culture (see below). Flag IP of the mixtures showed that the co-precipitated SsPG1-GFP gradually decreased with the increase of SsPINE1-GFP, and became undetectable when SsPINE1-GFP reached 10 μg. Instead, SsPINE1-GFP co-precipitated with AtPGIP1-3xFlag (Fig. 3b, left panel). The results suggest that

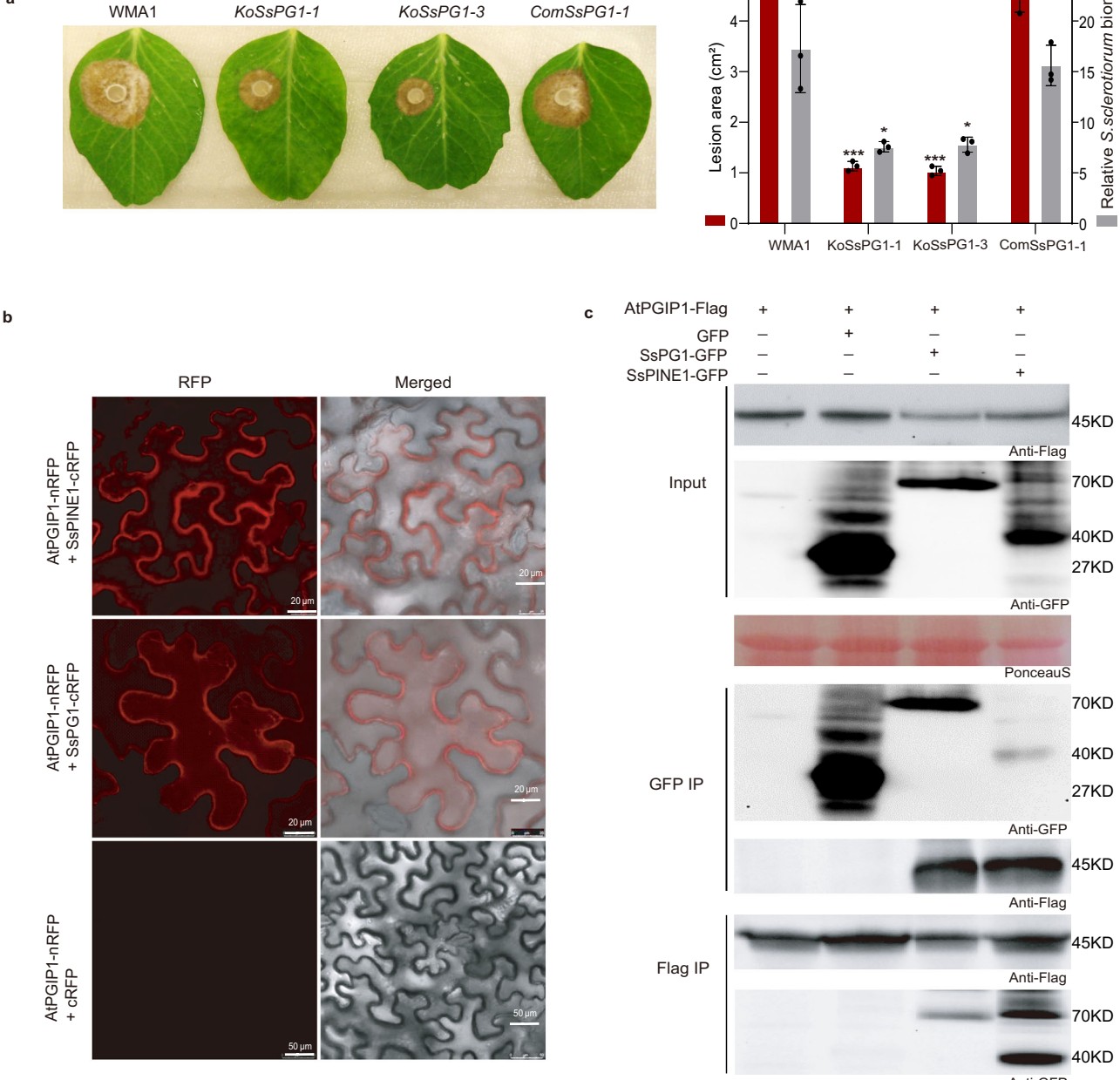

**Fig. 2 *Sclerotinia sclerotiorum* SsPG1 is critical for virulence and also interacts with AtPGIP1. a** Disease symptoms and disease quantification (lesion area and relative *S. sclerotiorum* biomass) on pea leaves caused by WMA1, two *SsPG1* mutants *KoSsPG1-1* and *KoSsPG1-3*, and the complement strain *ComSsPG1-1*. Representative leaves were photographed at 2 dpi. Data represent mean ± s.d. of $n = 3$ biological replicates. * and *** indicate significant difference from the wildtype WMA1 at $p < 0.05$ and 0.001, respectively, in two-tailed t-test. The experiment was performed three times independently with similar results obtained. **b** Bimolecular fluorescence complementation (BiFC) assay showed that SsPG1 as well as SsPINE1 interacts with AtPGIP1. *N. benthamiana* leaves were agroinfiltrated with a mixture of *Agrobacterium tumefaciens* strains harboring constructs AtPGIP1-nRFP and SsPG1-cRFP (top panel), AtPGIP1-nRFP and SsPINE1-cRFP (middle panel), and the negative control AtPGIP1-nRFP and cRFP (lower panel). The RFP fluorescence was monitored at 2 days post-agroinfiltration using confocal laser scanning microscope. Representative images are shown. The experiment was performed three times with similar results obtained. **c** Co-immunoprecipitation (Co-IP) assay confirmed that both SsPG1 and SsPINE1 were associated with AtPGIP1 *in planta*. Co-IP constructs indicated on the top were transiently expressed in *N. benthamiana* leaves. Immunoprecipitations were performed with anti-GFP agarose (GFP IP) or anti-Flag agarose (Flag IP). Western blot with anti-Flag antibody showed that AtPGIP1 was coprecipitated with SsPINE1 and also SsPG1 and Western blot using anti-GFP antibody showed that SsPG1 and SsPINE1 were coprecipitated with AtPGIP1. Representative images are shown. The experiments were performed three times independently with similar results obtained. Source data are provided as a Source data file.

SsPINE1 dissociated the SsPG1/AtPGIP1 complex and formed a complex with AtPGIP1.

In the reciprocal Flag IP assay, equal volumes of the same tobacco leaf lysate co-infiltrated with *SsPINE1-GFP* and *AtPGIP1-* *3xFlag* constructs were mixed with 0, 100 ng and 10 µg of the purified SsPG1-GFP. Addition of purified SsPG1-GFP protein to the leaf lysate did not reduce SsPINE1-GFP in the immunoprecipitates, and no detectable SsPG1-GFP was co-precipitated

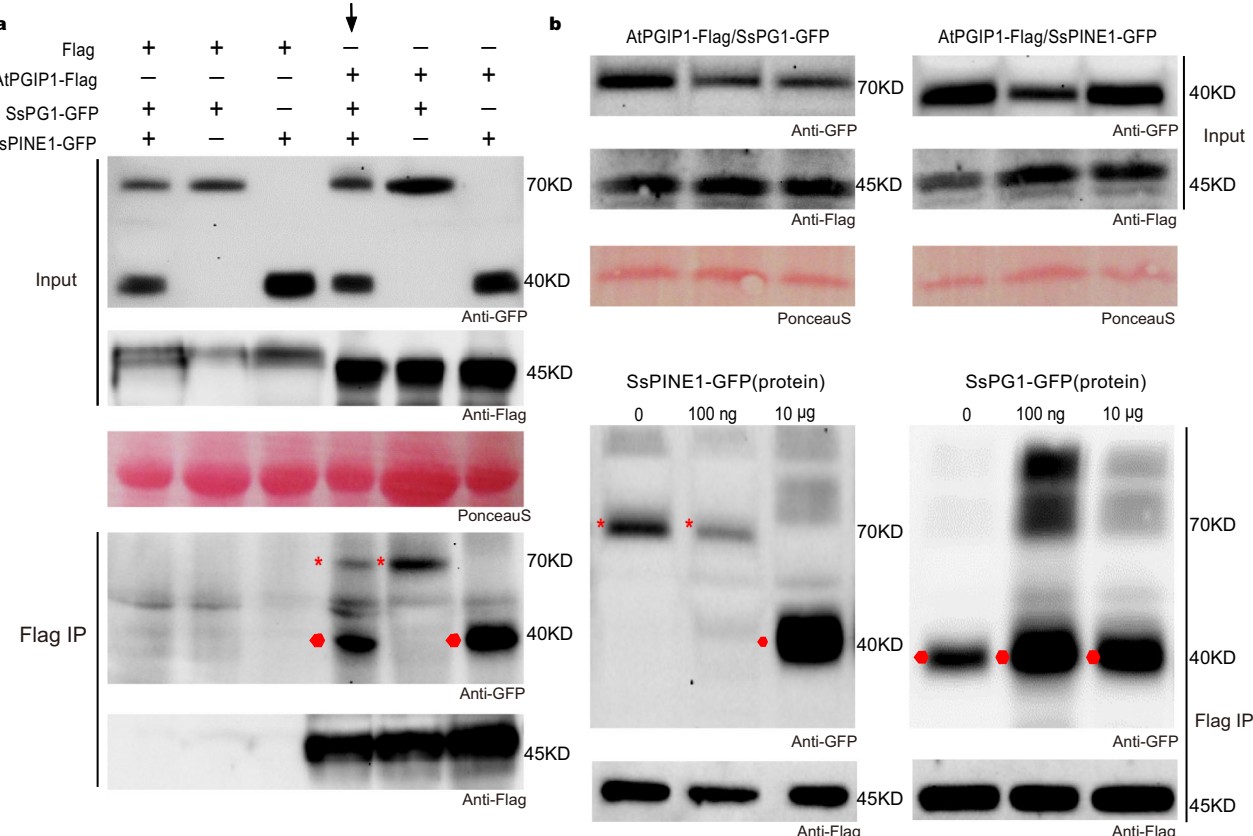

**Fig. 3 SsPINE1 outcompetes SsPG1 in binding with AtPGIP1. a** More SsPINE1 co-precipitated with AtPGIP1 than SsPG1 in co-IP assay. When both SsPINE1-GFP and SsPG1-GFP were together co-infiltrated with AtPGIP-3XFlag, SsPINE1-GFP co-precipitated more effectively with AtPGIP1-3xFlag than SsPG1-GFP (indicated by the arrow). **b** SsPINE1 enhances the dissociation of SsPG1 from AtPGIP1 (left panel). In contrast, SsPG1 cannot compete effectively with SsPINE1 for AtPGIP1 binding (right panel). *N. benthamiana* leaves were agroinfiltrated with *SsPG1-GFP* and *AtPGIP1-3xFlag*. 0, 100 ng and 100 μg of purified SsPINE1-GFP fusion protein from yeast culture was mixed with equal volumes of the same leaf lysate, immunoprecipitated with anti-Flag agarose (Flag IP), and analyzed by Western blot using anti-GFP antibody. Similarly, purified 0, 100 ng and 100 μg of SsPG1-GFP fusion protein from yeast culture were mixed with equal volumes of the same leaf lysate of *N. benthamiana* leaves agroinfiltrated with *SsPINE1-GFP* and *AtPGIP1-3xFlag*. Red asterisks and round dots indicate the expected sizes of SsPG1-GFP and SsPINE1-GFP, respectively. Representative results are shown, and each experiment was performed three times with similar results obtained. Source data are provided as a Source data file.

with AtPGIP1 (Fig. 3b, right panel). The results showed that SsPG1 cannot replace SsPINE1 in the SsPINE1/AtPGIP1 complex. These results of the modified Co-IP assays suggest that SsPINE1 inactivates AtPGIP1 through dissociating SsPG1 from AtPGIP1, consequently enhancing PG activity.

**SsPINE1 nullifies the PG inhibitory effect of AtPGIP1.** Functional interactions among SsPG1, AtPGIP1 and SsPINE1 were tested by enzymatic assays. AtPGIP1-His, SsPINE1-GFP-His and SsPG1-GFP-His fusion proteins were purified from the yeast *Pichia pastoris* expression system (Supplementary Fig. 6). As expected, only SsPG1-GFP-His showed PG activity in an agar diffusion assay (Supplementary Fig. 7a) and the PG activity increased as SsPG1-GFP protein concentration increased (Fig. 4a). The PG activity of SsPG1-GFP-His was inhibited by addition of AtPGIP1-His (Fig. 4a). Addition of purified SsPINE1-GFP protein suppressed the PG inhibitory effect of AtPGIP1 and restored the PG activity of SsPG1 (Fig. 4a). Such dynamics of the PG inhibition by AtPGIP1 and counter-inhibition by SsPINE1 were captured by monitoring the accumulation of D-galacturonic acid (Fig. 4b). The proper secondary structure of AtPGIP1 is important for its function because boiled AtPGIP1-His lost its inhibitory effect (Supplementary Fig. 7b). Since SsPINE1-GFP itself did not show PG activity (Supplementary Fig. 7a), the

increase of the PG activity in the SsPG1-GFP /AtPGIP1 mixture must be through negating the PG inhibitory effect of AtPGIP1.

The purified AtPGIP1 protein not only inhibited SsPG1 activity in hydrolyzing polygalacturonic acid in enzyme assays but also inhibited *Sclerotinia* infection of plant (Fig. 4c) in pathogenicity bioassays. Application of purified AtPGIP1-His protein reduced *Sclerotinia* infection. In particular, the ΔSsPINE1 mutant was completely inhibited from initiating infection by the applied AtPGIP1. Disease lesion size caused by the wildtype strain was also significantly reduced by application of AtPGIP1, although production of SsPINE1 allowed the wildtype strain to initiate infection even in the presence of excess AtPGIP1 (Fig. 4c, d), indicating that SsPINE1 is functional in inactivating AtPGIP1 to retain virulence.

**The β-sheet P2 region of AtPGIP1 determines binding specificity.** Previous research showed that PGIP binding affinity is influenced by amino acid residues on the β-sheet-formed concave surface, which determines the specificity for PGs[40,41]. Thus, we dissected this concave surface by conducting directed mutagenesis on three regions creating three AtPGIP1 mutation variants P1, P2 and P3, as illustrated in Supplementary Fig. 8a. All three mutation variants of AtPGIP1 were shown to have folding structure similar to the native AtPGIP1 (Supplementary Fig. 8b). The mutation variants were fused with a His-tag and expressed in

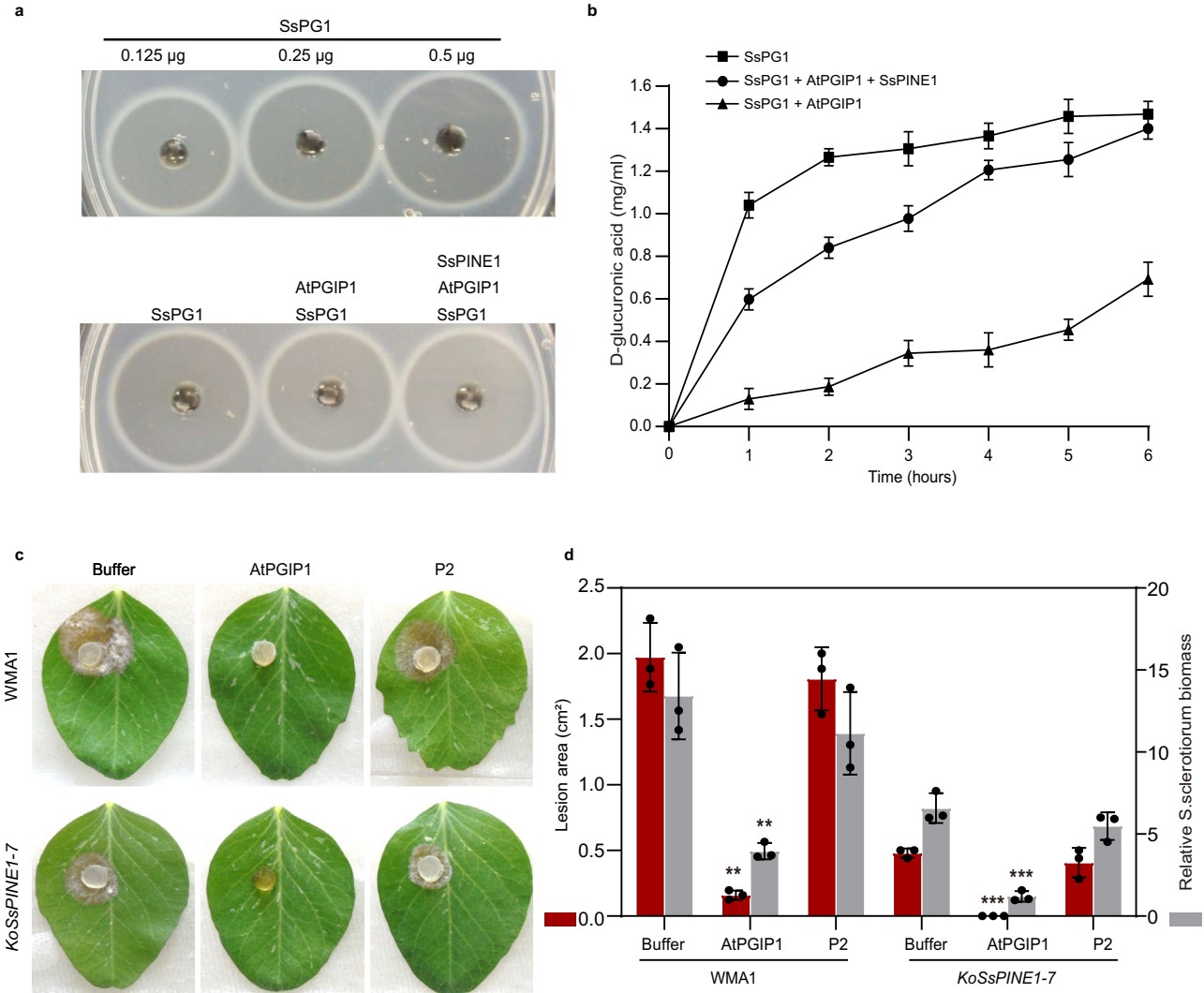

**Fig. 4 AtPGIP1 inhibits SsPG1 hydrolytic activity and Sclerotinia infection, whereas SsPINE1 nullifies AtPGIP inhibition. a** PG activity (ring diameter) is proportional to SsPG1 concentration (top panel), and addition of AtPGIP1 (10 μg) to SsPG1 (0.5 μg) reduced its PG activity and further addition of SsPINE1 (10 μg) negated the inhibition effect of AtPGIP1 and restored PG activity of SsPG1 (bottom panel). **b** PG activity of SsPG1 (0.5 μg) alone, in the presence of AtPGIP1 (10 μg) or in the presence of both AtPGIP1 and SsPINE1 (10 μg each) was determined using the DNSA method by measuring accumulation of D-galacturonic acid. Data represent means ± s.d. of *n* = 3 independent replicates. **c** Purified AtPGIP1 protein inhibited *Sclerotinia* infection. AtPGIP1 protein (10 μg) and its mutation variant AtPGIP1-P2 (10 μg; see Fig. 5) were applied to *Sclerotinia sclerotiorum* inoculum of wildtype strain WMA1 and *SsPINE1*-deletion mutant *KoSsPINE1-7* in detached pea leaf bioassays with PBS buffer as a control for comparison. Representative photos (2 dpi) are presented. **d**. Lesion area and relative biomass of *S. sclerotiorum* wildtype strain WMA1 and *SsPINE1*-deletion mutant *KoSsPINE1-7* in the presence of AtPGIP1 (10 μg), AtPGIP1-P2 (10 μg) or PBS buffer 2 dpi. Biomass was measured by genomic DNA quantitative PCR. Data represent means ±s.d. with *n* = 3 biological replications. Treatment means are significantly different from that of the PBS buffer control at *P* < 0.01 (**) or 0.001 (***), respectively, in two-tailed *t* test. The experiment was performed twice with similar results obtained. Source data are provided as a Source data file.

yeast. Purified variant P1, P2, P3 and native AtPGIP1 proteins were verified by Western blot using the anti-His antibody (Supplementary Fig. 8c). The differential contributions of P1, P2 and P3 regions to AtPGIP1 binding specificity were tested using Co-IP assays (Fig. 5a, b). *N. benthamiana* leaves were co-infiltrated with SsPINE1-GFP expression construct and construct expressing 3xFlag-tagged AtPGIP1, AtPGIP1-P1, AtPGIP1-P2, or AtPGIP1-P3. Flag IPs demonstrated that AtPGIP1-P2 lost SsPINE1 binding ability, and AtPIGP1-P3 also had reduced binding with SsPINE1 (Fig. 5a). Similarly, Flag Co-IPs using SsPG1-GFP confirmed that AtPGIP1-P2 also plays a primary role in AtPGIP1 binding with SsPG1 (Fig. 5b). These results above support a working model whereby SsPINE1 binds the P2 region to suppress the SsPG1-inhibition activity of AtPGIP1, and thereby facilitates SsPG1-mediated plant cell wall degradation. The inhibitory activities of the mutation variants were compared with that of the wildtype AtPGIP1 using an agarose diffusion assay. Results showed that the P1 variant had the same level of inhibitory effect on SsPG1 as the wildtype AtPGIP1(Fig. 5c, d), suggesting that the P1 region is not involved in binding, which is consistent with the Co-IP results. In contrast, P2 and P3 mutations reduced the PG inhibitory capacity of AtPGIP1, as shown in the agar diffusion assay as well as in the accumulated D-galacturonic acid (Fig. 5c, d). Consistent with the enzymatic assays, the AtPGIP1-P2 variant protein also lost its ability to inhibit *Sclerotinia* infection in a leaf infection bioassay (Fig. 4c). These results of Co-IP binding assay, enzyme activity assays and leaf infection bioassay suggest that both P2 and P3 regions are involved in AtPGIP1-mediated inhibition of SsPG1, with the P2 region playing a more critical role.

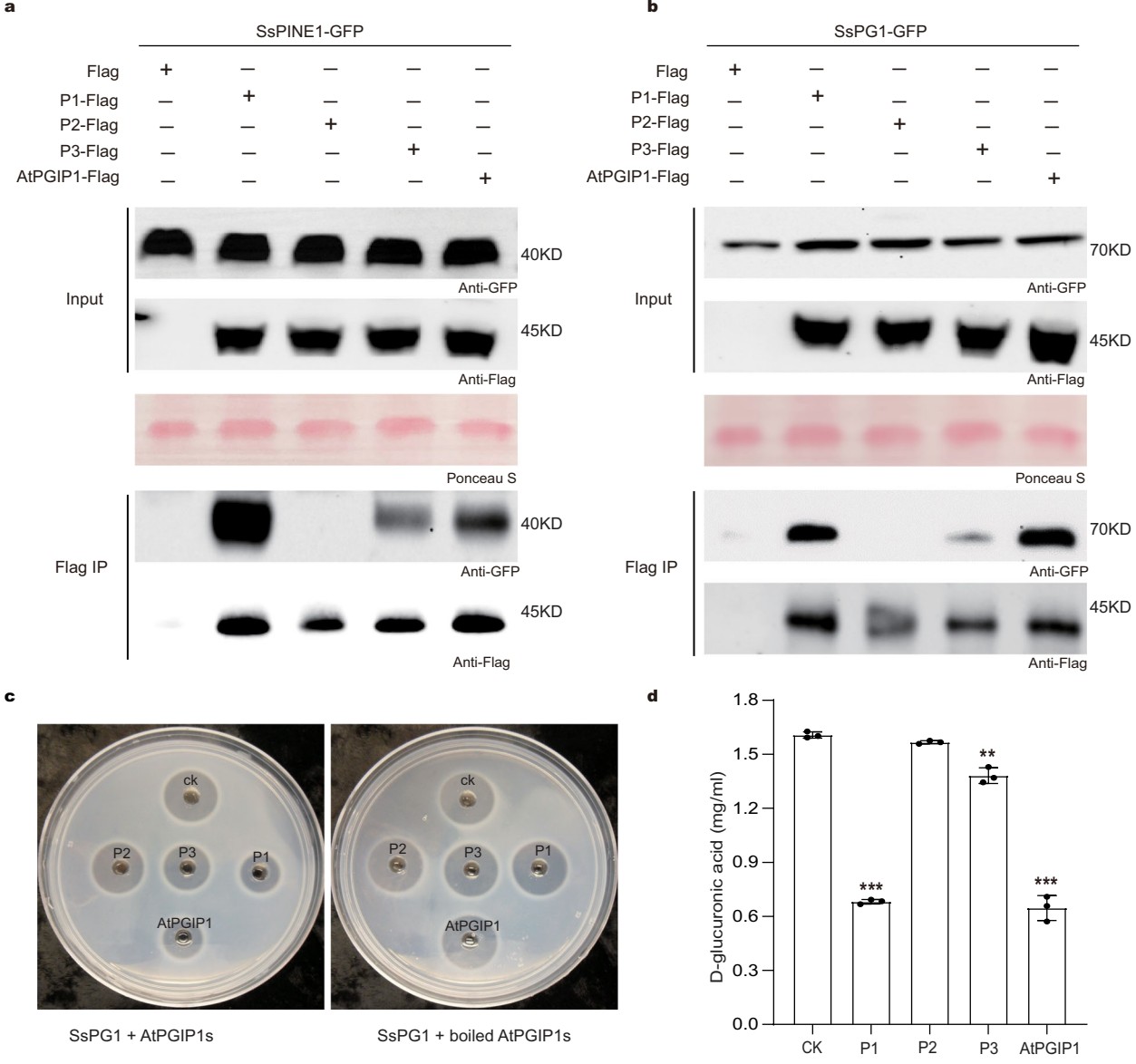

**Fig. 5 The P2 region of AtPGIP1 determines its binding specificity with SsPINE1 and SsPG1. a** Co-IP assays showing the AtPGIP1 variants P2 and P3 have lost and reduced, respectively, affinity with SsPINE1. SsPINE1-GFP was co-expressed with 3xFlag-tagged AtPGIP1, AtPGIP1-P1, AtPGIP1-P2, or AtPGIP1-P3 in *N. benthamiana* leaves. The samples were then used for Flag IPs and anti-GFP Western blots. **b** Similar Co-IP assays were carried out to test the bindings of SsPG1-GFP to 3xFlag-tagged AtPGIP1, AtPGIP1-P1, AtPGIP1-P2, or AtPGIP1-P3. Co-IP assays showing the AtPGIP1 variants P2 and P3 have lost and reduced affinity with SsPG1, respectively. SsPG1-GFP was co-expressed with 3xFlag-tagged AtPGIP1, AtPGIP1-P1, AtPGIP1-P2, or AtPGIP1-P3 in *N. benthamiana* leaves. The samples were then used for Flag IPs and anti-GFP Western blots. **c** Agar diffusion assay showing the SsPG1-inhibitory activity of AtPGIP1 and its mutation variants P1, P2 and P3. The wells were applied with 0.5 μg of SsPG1 alone (CK) and with 10 μg of either variant P1, P2, P3, or wildtype AtPGIP1 proteins. **d** PG activity of SsPG1 alone (CK) and in the presence of AtPGIP1 or its mutation variants were determined by measuring the released D-galacturonic acid. Data represent means ± s.d. of $n = 3$ independent replicates. ** and *** indicate significant difference from the check (CK, SsPG1 alone) at $p < 0.01$ and 0.001, respectively, in two-tailed t-test. Source data are provided as a Source data file.

**SsPINE1 lowers plant immunity.** Stable 35S::*SsPINE1-GFP* and 35S::*AtPGIP1-3xFlag Arabidopsis* (Col-0) overexpressing lines were created to investigate *in planta* the roles of SsPINE1 and AtPGIP1 in plant-*Sclerotinia* interactions. Expression of *SsPINE1-GFP* in *A. thaliana* was confirmed by Western blot and fluorescence microscopy (Supplementary Fig. 9a, b), and overexpression of *AtPGIP1-3xFlag* was confirmed by Western blot and RT-qPCR (Supplementary Fig. 9c, d). The AtPGIP-3xFlag protein was detected in the apoplastic fluid (Supplementary Fig. 9e), suggesting that AtPGIP1 is also a secreted protein. All transgenic lines showed normal growth and morphology (Supplementary Fig. 9f).

The three genotypes (Col-0, 35S::SsPINE1 and 35S::AtPGIP1) of *A. thaliana* plants were tested for their reactions to infection by three genotypes of *S. sclerotiorum* (wild-type, ΔSsPINE1 and ΔSsPG1) strains. Deletion of either *SsPINE1* or *SsPG1* significantly reduced *S. sclerotiorum* virulence (Fig. 6), with the disease lesion sizes on Col-0 plants reduced by around 30% and 95%, respectively. In general, ectopic expression of *SsPINE1* increased *A. thaliana* susceptibility to all three genotypes of *S. sclerotiorum* (Fig. 6a, b). Compared to wild-type *A. thaliana* (Col-0), plants expressing *SsPINE1* (35S::*SsPINE1*) exhibited significantly increased disease lesion areas and relative *Sclerotinia* biomass upon *S. sclerotiorum* (WMA1, ΔSsPINE1 or ΔSsPG1) infection

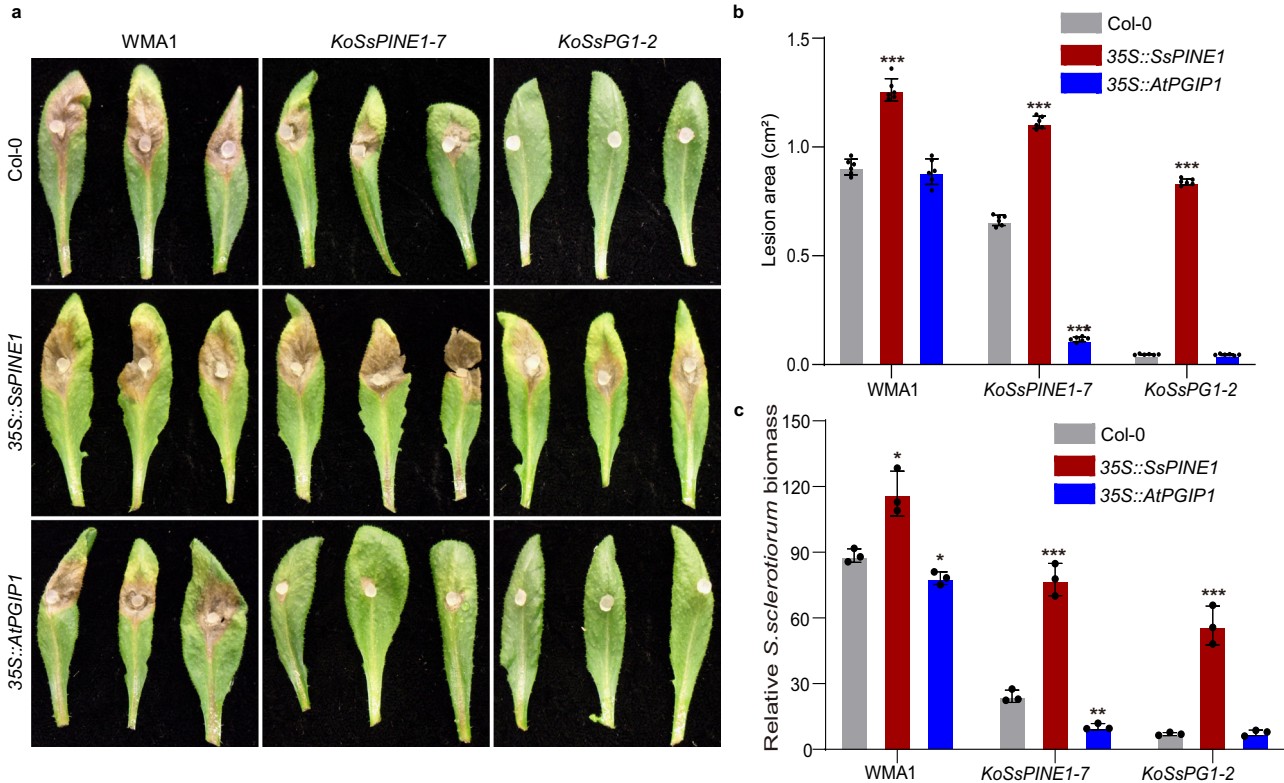

**Fig. 6 Expression of *SsPINE1* and *AtPGIP1* in *Arabidopsis* altered plant response to *Sclerotinia sclerotiorum* infection. a** *Arabidopsis* wild type Col-0, *SsPINE1-GFP*-expressing (35S::SsPINE1) and *AtPGIP1-3xFlag*-overexpressing (35S::AtPGIP1) lines were challenged with *S. sclerotiorum* WMA1, *KOSsPINE1-7* and *KOSsPG1-2*. Representative leaves were photographed 48 h post-inoculation. The experiment was repeated three times with similar results obtained. **b** Lesion area caused by WMA1, *KOSsPINE1-7* and *KOSsPG1-3* on Col-0, 35S::SsPINE1 and 35S::AtPGIP1 *Arabidopsis* lines at 48 hpi. Data represent means ± s.d. with $n = 6$ replicates, Treatment means are significantly different from that of the wildtype Col-0 at $P < 0.001$ (***) in two-stailed $t$ tests. **c** Relative *S. sclerotiorum* biomass of WMA1, *KOSsPINE1-7* and *KOSsPG1-3* on Col-0, 35S::SsPINE1 and 35S::AtPGIP1 *Arabidopsis* lines at 48 hpi, measured by genomic DNA quantitative PCR. Data represent means ± s.d. with $n = 3$ replicates, Treatment means are significantly different from that of the wildtype Col-0 at $P < 0.05$ (*), 0.01 (**) or 0.001 (***), respectively, in two-tailed $t$ tests. The experiment was performed three times with similar results obtained. Source data are provided as a Source data file.

(Fig. 6b, c). The most dramatic increase in lesion area and relative *Sclerotinia* biomass was observed when inoculated with the ΔSsPG1 mutant (Fig. 6b, c), which suggests that overexpression of *SsPINE1* in the host plant can aid *Sclerotinia* virulence factors other than SsPG1, likely other SsPGs, suggesting that AtPGIP1 may inhibit other SsPGs in addition to SsPG1. Notably, infection levels on the *SsPINE1*-expressing plants infected by the ΔSsPINE1 mutant strain were higher than that on the wild-type Col-0 plants infected by the wild-type *S. sclerotiorum* (Fig. 6), suggesting that ectopic expression of *SsPINE1* in the host can adequately restore virulence of the ΔSsPINE1 mutant beyond the wild-type level.

On the other hand, overexpression of *AtPGIP1* generally enhanced plant immunity, particularly to the ΔSsPINE1 mutant (Fig. 6). For the ΔSsPG1 mutant that possesses minimum virulence on the wildtype Col-0 plant, no further reduction in virulence is observed in the AtPGIP1-overexpression plant. For the wildtype WMA1 strain, such enhanced resistance was not obvious in terms of lesion area Fig. 6a, b). This observation is consistent with previous reports that the high susceptibility of *A. thaliana* makes it difficult to differentiate genotypic differences in response to *S. sclerotiorum* infection[42,43]. However, overexpression of AtPGIP1 did show further reduced pathogen biomass measured by relative DNA content compared to wildtype Col-0 (Fig. 6c). Such a clear enhanced resistance by PGIP-overexpression to the ΔSsPINE1 mutant but only subtle difference to the wild-type strain further demonstrates the critical role of *SsPINE1* in *Sclerotinia* virulence. The presence of SsPINE1 in the

wildtype *Sclerotinia* strain is sufficient to inactivate the extra AtPGIP1 in the *AtPGIP1*-overexpressing plants and minimize immunity enhancement effects of PGIP's. Previous studies showed that purified *Sclerotinia* PGs were very sensitive to inhibition by plant PGIPs[19,44], but such sensitivity is lost in bioassays since *S. sclerotiorum* can readily infect and macerate host tissue, even in PGIP-overexpressing plants[19,44], which may be at least partially due to the PGIP inactivation by endogenous SsPINE1.

**Botrytis cinerea BcPINE1, a SsPINE1 homolog, contributes to virulence and also binds to AtPGIP1.** SsPINE1 is predicted to have an internal repeat structure (Supplementary Fig. 10a). A protein BLAST search of the NCBI protein database[45] using the SsPINE1 amino acid sequence as query resulted in identification of 26 accessions with high percent identity distributed in four fungal genera (*Sclerotinia*, *Botrytis*, *Monilinia* and *Rustroemia*). The internal repeat structure is also conserved in the SsPINE1 homologs (Supplementary Fig. 10b). Six accessions representing the four genera were selected for phylogenetic analysis. SsPINE1 is closely related to its homolog in *Botrytis cinerea* (Supplementary Fig. 10c), another notorious broad host range necrotrophic plant pathogen[8].

To investigate whether SsPINE1-mediated PGIP inactivation is a conserved mechanism in broad host range necrotrophic fungi, we generated two knockout mutants (*KoBcPINE1-3* and

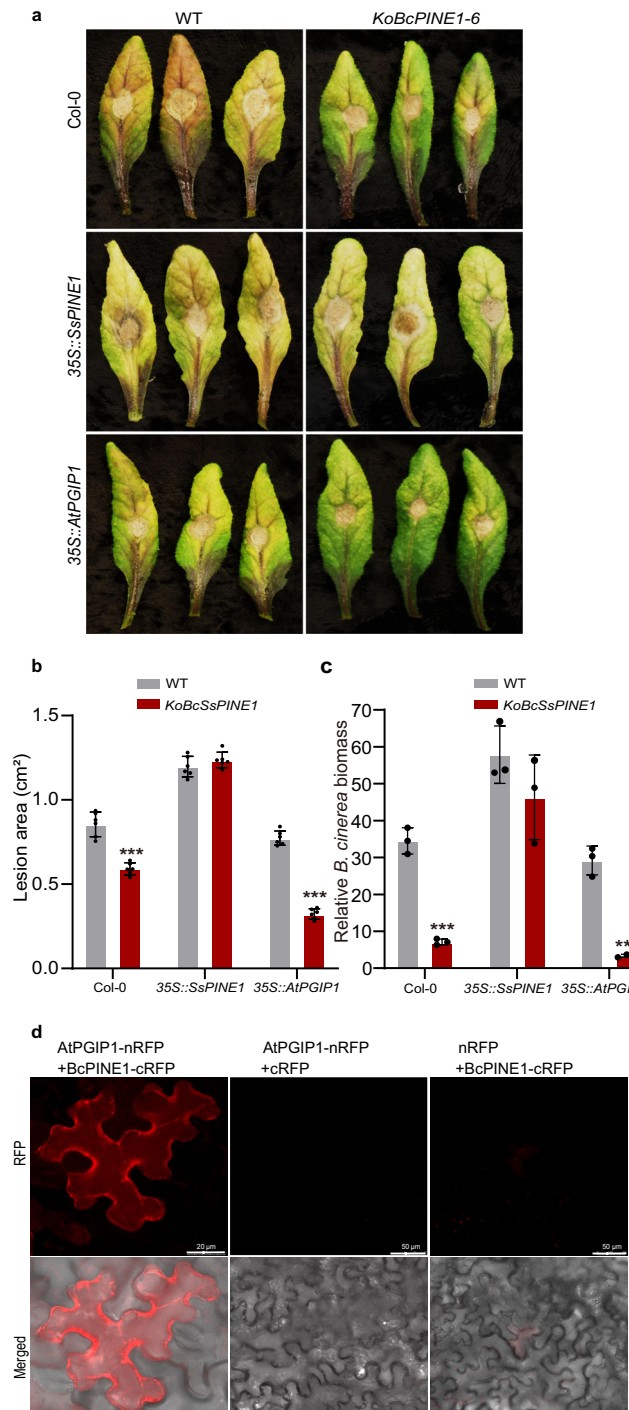

**Fig. 7 *Botrytis cinerea* BcPINE1 contributes to virulence and also binds to AtPGIP1. a** Deletion of *BcPINE1* reduced virulence of *B. cinerea* on Arabidopsis lines. *Arabidopsis* wild type Col-0, *SsPINE1-GFP*-expressing (35S::SsPINE1) and *AtPGIP1-3XFlag*-overexpressing (35S::AtPGIP1) lines were assayed with *Botrytis cinerea* wildtype B05.10 and Δ*BcPINE1* mutant *KoBcPINE1-6* strains. Representative leaves were imaged 48 h post-inoculation. The experiment was repeated three times with similar results. **b** Lesion area caused by B05.10 and *KoBcPINE1-6* strains of *Bortytis cinerea* on Col-0, *35S::SsPINE1* and *35S::AtPGIP1 Arabidopsis* lines at 48 hpi. Data were means ±s.d. with *n* = 6 replicates. Treatment means are significantly different from that of the wildtype Col-0 at *P* < 0.001 (***) in two-tailed *t* tests. The experiment was performed three times with similar results. **c** Relative *B. cinerea* biomass of *Bortytis cinerea* B05.10 and *KoBcPINE1-6* strains on Col-0, *35S::SsPINE1* and *35S::AtPGIP1 Arabidopsis* lines at 48 hpi, measured by genomic DNA quantitative PCR. Treatment means are significantly different from that of the wildtype Col-0 at *P* < 0.01 (***) in two-tailed *t* tests. The experiment was performed three times with similar results. **d** Bimolecular fluorescence complementation (BiFC) confirmed that BcPINE1 interacts with AtPGIP1. *N. benthamiana* leaves were agroinfiltrated with a mixture of *Agrobacterium tumefaciens* strains harboring constructs AtPGIP1-nRFP and BcPINE1-cRFP and negative controls AtPGIP1-nRFP and cRFP, and nRFP and BcPINE1-cRFP. The RFP fluorescence was monitored at 2 days post-agroinfiltration using confocal laser scanning microscopy. Representative images are displayed. The experiment was performed three times with similar results. Source data are provided as a Source data file.

was significant only in terms of relative *B. cinerea* biomass (Fig. 7b, c). Physical interaction between BcPINE1 and AtPGIP1 was demonstrated using BiFC assays (Fig. 7d). These results showed that as an SsPINE1 homolog in *B. cinerea*, BcPINE1 has similar functions of inactivating PGIP activity through direct binding.

## Discussion

Our results present five lines of evidence that SsPINE1 plays important roles in necrotrophic virulence through inactivating plant PGIP: First, deletion of *SsPINE1* significantly reduced virulence; Second, SsPINE1 is a secreted protein and specifically binds with AtPGIP1 and is capable of replacing SsPG1 from preformed SsPG1-AtPGIP1 complex. Third, SsPINE1 can negate the inhibitory effect of AtPGIP1 on polygalacturonase. Forth, heterologous expression of *SsPINE1* in plant lowered plant immunity. And fifth, PINE1 is conserved among many broad host range necrotrophic pathogens.

The battle between fungal PGs and plant PGIPs has been well documented since the discovery of PGIP in 1971 and has been thought to be one-on-one engagement[12,13,15]. PGs from *S. sclerotiorum* and other fungi play important roles in inducing necrosis and degrading pectin of plant cell walls in macerating host tissue and releasing nutrients for fungal growth[10,11,46]. Plants have evolved PGIPs in defense. PGIPs can specifically bind to fungal PGs and modulate PG activity, in the process slowing down pectin degradation and allowing accumulating oligogalacturonides that act as elicitors of plant defense response[12,13], and also blocking the ability of PGs' to induce necrosis[46]. Both fungal PGs and plant PGIPs have well conserved primary and secondary structures required for the interaction. Purified *S. sclerotiorum* PGs are very sensitive to PGIP inhibition[19,44]. *S. sclerotiorum* can cause rapid tissue maceration on a wide range of host plants including virtually all dicots and pectin-rich monocots, even in some transgenic plants with elevated PGIP expression[19,20]. Here we introduce a third player—SsPINE1—in the PG-PGIP battleground. SsPINE1 contributes to *S. sclerotiorum*'s exceptional ability to macerate host tissue by inactivating PGIPs and

*KoBcPINE1-6*) of *BcPINE1* (BC1G_04506) of *Botrytis cinerea* wildtype strain B05.10 (Supplementary Fig. 11a). The Δ*BcPINE1* mutants *KoBcPINE1-3* and *KoBcPINE1-6* showed similar growth rate and colony morphology to the wildtype strain B05.10 (Supplementary Fig. 11b). The Δ*BcPINE1* mutant showed reduced virulence to wild-type *A. thaliana*. Expressing *SsPINE1* in *A. thaliana* increased susceptibility to both *B. cinerea* wildtype and Δ*BcPINE1* mutant (Fig. 7), suggesting expression of *SsPINE1* in the plant can restore virulence of the Δ*BcPINE1* mutant of *B. cinerea* to the wildtype level. Similar to the situation with *S. sclerotiorum*, overexpression of AtPGIP1 enhanced resistance to *B. cinerea*, particularly to the BcPINE1-knockout mutant *KoBcPINE1-6*. The enhanced resistance to the wildtype *B. cinerea*

protecting PG activity. Albersheim and Anderson[15] predicted that potential PGIP inhibitors should have higher affinity with PGIPs and/or be present at higher abundance to be functionally effective. Here, we identified for the first time the virulence effector protein SsPINE1 as a PGIP inactivator. SsPINE1 can dissociate SsPG1 from binding with AtPGIP1, nullifying AtPGIP1 inhibition and enhancing PG activity. SsPINE1 targets a plant PGIP which is highly conserved; thus, SsPINE1 contributes to broad spectrum of virulence, consistent with the wide host range of *S. sclerotiorum*. Furthermore, SsPINE1 homologs were found in species belonging to three other fungal genera: *Botrytis*, *Monilinia* and *Rutstremia* of the Ascomycota order Helotiales (Supplementary Fig. 8). BcPINE1, a homolog of SsPINE1 in *B. cinerea*, also interacts with AtPGIP1 and contributes to virulence, suggesting that SsPINE1-mediated virulence mechanism is conserved among necrotrophic fungi of the order *Helotiales* under the selection pressure exerted by host PGIPs. The discovery of SsPINE1, a counter-inhibitor of plant PGIPs, has uncovered a mechanism of fungal necrotrophy. The involvement of SsPINE1 in the PG-PGIP interaction makes this tripartite battle more intricate, since PGIPs are known to have other functions besides inhibiting fungal PGs[47,48]. The potential effect of SsPINE1 on the other functions of PGIPs remains to be explored.

The genomes of both *S. sclerotiorum* and *B. cinerea* encode at least five endo-PGs[8]. The critical role of the PGs in virulence of *B. cinerea* has been shown in genetic analysis through gene-deletion[9]. However, in *S. sclerotiorum*, the importance of PGs was shown through enzyme analysis and gene expression patterns[11,38]. Here for the first time we showed that deletion of a *Sclerotinia* PG gene (SsPG1) significantly reduced virulence. SsPG1 is a neutral PG and is constitutively expressed and its expression is significantly induced upon inoculation onto plants[11]. Interestingly, in the SsPG1-deletion mutants, no obvious compensation by other SsPGs was observed (Fig. 2a), demonstrating the critical role of SsPG1 in initiating infection. However, the SsPG1-deletion mutant did cause significant necrotic lesions on SsPINE1-expressing plants, suggesting necrotrophic activities of other SsPGs when the PG-inhibition exerted by plant PGIP is mitigated by SsPINE1 (Fig. 6).

Heterologous expression of virulence factors in plants is often used to prove the roles of the effector in virulence. Overexpression of SsPINE1 in *A. thaliana* significantly increased plant susceptibility to *Sclerotinia* infection. Presence of SsPINE1 in the transgenic plant can compensate for the lack of SsPINE1 in the SsPINE1-deletion mutant and restore its virulence to the wildtype level, similar to the effect of complementing the SsPINE1-deletion in the mutant.

The ten conserved LRR modules of AtPGIP form the concave surface that is required for interaction with fungal PGs[35,40]. We suspected that SsPINE1 also interacts with PGIPs on the concave surface. In order to determine approximate region of the concave surface that interacts with SsPINE1, we created three variant proteins through amino acid substitutions. Co-IP experiments showed that the interaction occurs in the middle region because SsPINE1 did not coprecipitate with mutation variant P2. Apparently, SsPG1 also binds to the same region. These results provide information for more precise dissection of these regions to explore the potential of engineering PGIPs that can escape recognition of SsPINE1 and still retain PG binding and inhibition.

Current efforts in deploying PGIPs for improving plant resistance focus on enhancing the expression of known PGIPs and discovering or engineering novel PGIPs with enhanced potency and a broadened PG recognition spectrum[13,49]. Our results reveal SsPINE1 acts in the PG-PGIP battleground. *S. sclerotiorum* produces SsPINE1 that disables plant PGIPs, which results in the

release of PGs and the enhancement of necrotrophic virulence. Homologs with similar functions are also present in *B. cinerea* and in other necrotrophic pathogens. Now with the knowledge of PGIP counter-inhibitors such as SsPINE1, newly engineered PGIPs that can avoid the recognition by SsPINE1 could be deployed for effective breeding of resistance plants against PINE1-possessing necrotrophic fungal pathogens.

## Methods

**Bacterial and fungal strains, growth conditions and DNA analysis.** The wild-type strain WMA1 (ATCC MYA-4521) of *Sclerotinia sclerotiorum* was used in this study[28,50] and routinely cultured on PDA (Difco Laboratories, Detroit) or on SY medium (5 g sucrose and 0.5 g yeast extract and 15 g agar per L). Knockout mutants of the *SsPINE1* (SS1G_08128) and *SsPG1* (SS1G_10167) genes were maintained on PDA. Complemented mutants were cultured on PDA plates amended with G418 at 100 µg/ml. Yeast *Pichia pastorisi* GS115 strain was from Invitrogen. Bacterial strains *Escherichia coli* JM109 and *Agrobacterium tumefaciens* GV3101 were cultured on LB medium. Gel electrophoresis, restriction enzyme digestion, gel blot and sequencing were performed using standard procedures[51].

**Gene replacement and complementation of *S. sclerotiorum*.** Gene replacement mutants for the *SsPINE1* and *SsPG1* genes of *S. sclerotiorum* were generated using a split marker technique[28,52]. About 500 bps each of the 5′ and 3′ flanking sequences of *SsPINE1* or *SsPG1* open reading frames (ORF) were amplified from genomic DNA of the wildtype strain WMA1 with appropriate primer pairs (Supplementary Table 2). The PCR products were then cloned into the upstream and downstream regions of the *hyg* cassette in the vector pCH-3300, using the Gibson Assembly Master Mix kit (New England Biolabs, MA, USA) according to the manufacturer's instructions to generate plasmids pCH-PINE1 and pCH-PG1, respectively (Supplementary Table 3). Two overlapping DNA fragments 5′-HY and YG-3′ with truncated hygromycin-resistant *hyg* gene sequence were amplified with primers (E1-Hind-L/HY-R and YG-F/E1-Knp1-R from vector pCH-PINE1 for *SsPINE1* knockout; PG1-Hind-L/HY-R and YG-F/PG1-Knp1-R from vector pCH-PG1 for *SsPG1* knockout, respectively (Supplementary Table 2). Purified PCR products of the 5′-HY and YG-3′ fragments of the respective knockout constructs were mixed in equal molar quantities and used to transform protoplasts of WMA1 generated using lysing enzymes from *Trichoderma harzianum* (L1412, Sigma Aldrich). Because the *Sclerotinia* mycelium is multinucleate (up to 250 nuclei in a cell) and lacks a conidial state, special caution was taken in purifying the gene knockout transformants. Putative transformants that survived hygromycin selection in the transformation were initially purified through three rounds of hyphal tipping and the proper insertion of the *hyg* gene was verified by PCR using a flanking primer located outside the knockout construct (E1YZ-L-1 for *SsPINE1* and PGYZ-L for *SsPG1*) and another primer (HY-R) located in the *hyg* gene. Once the correct location of the *hyg* gene was confirmed, the transformants were further purified by up to 20 rounds of hyphal tipping until the deletion allele is in homokaryotic state (e.g. absence of the wildtype allele in the genome by detection with RT-PCR and Southern blot analysis).

To complement the gene knockout mutants, the full-length wildtype allele of the *SsPINE1* and *SsPG1* genes without stop codon was amplified from cDNA using RT-PCR with appropriate primer pairs (Sac1CE1-L/ Kpn1CE1-R for complementing *SsPINE1* and PG1-Sac1/PG1-SMA1 for complementing *SsPG1*; Supplementary Table 2). Amplicons were fused with GFP in the Sac1/ Kpn1 sites of the vector pCETNS which contained the constitutive P*trpC* promoter and T*trpC* terminator, resulting in complementation vectors pCETNSE1 and pCETNSPG1, respectively (Supplementary Table 3). The complementation vectors were used to transform and complement the respective knockout mutants using *Agrobacterium*-mediated transformation[50]. Transformants were selected on PDA plates containing 150 µg/ml geneticin (G418; Ameresco, Ohio, USA) for two rounds of hyphal tipping under geneticin selection and the presence and expression of the wildtype alleles were further confirmed by Southern blot and RT-PCR analysis. The complementation vectors were also used to transform the wildtype strain to generate over-expression mutants for testing secretion of SsPINE1 and SsPG1.

**Characterization of *S. sclerotiorum* transformants.** *S. sclerotiorum* transformants (ΔSsPINE1 and ΔSsPG1 mutants and respective complement strains) were characterized for growth rate, colony morphology and pathogenicity. To assay growth rates, the wild type strain, deletion mutants (ΔSsPINE1 and ΔSsPG1) and complement transformants (SsPINE1 and SsPG1) were inoculated on to the center of PDA plates with three replicate plates using 4-mm diam agar discs taken from the edge of actively growing colonies of 2-day old cultures on PDA and incubated at 22 °C. Colony diameters were measured daily and colony morphology of these strains was examined after 10 d at 22 °C. Growth rate of SsPG1-knockout mutants was also compared with that of the wildtype on medium with polygalacturonic acid (2%) as the carbon source.

Pathogenicity assays were performed with detached leaves of 4-week old pea (*Pisum sativum* cv. Guido) plants[50]. Agar discs (4 mm diam.) from the edge of actively growing colonies of 2-day old cultures of the tested strains on PDA

medium at 22 °C were placed on the leaves with the mycelial side in contact with the leaves and incubated at room temperature (22–24 °C). Virulence was quantified using two methods: macroscopically measuring lesion diameters with a caliper and molecularly measuring the ratios of *S. sclerotiorum* DNA to pea DNA in infected leaves at 48 hpi. Lesion areas were calculated based on the average of two perpendicular measurements of the lesion diameter minus the agar plug area. Relative pathogen biomass was based on ratios of pathogen DNA to host plant DNA via genomic DNA quantitative PCR, for which the *S. sclerotiorum Actin* gene (primers in Supplementary Table 2) and pea *PsGAPDH* gene (primers in Supplementary Table 2) were amplified. Each treatment had three replicates (three leaves) and each experiment was performed three times.

**Secretion assay**. To test secretion of SsPINE1 and SsPG1, the complemented strains and overexpression strains (WMA1-ΔSsPINE1-SsPINE1-GFP; WMA1-SsPINE1-GFP; WMA1-ΔSsPG1-SsPG1-GFP; WMA1-SsPG1-GFP) were cultured in potato dextrose broth for 5 days at room temperature (22–24 °C). The culture filtrates were filtered with 0.22 μm Minisart non-pyrogenic filter to eliminate residual mycelia, and frozen at –80 °C and dried in the lyophilizer (Christ Delta 1–24 LSC Freeze Dryer) overnight. Lyophilized protein samples were dissolved in 250 μl phosphate buffered saline (PBS) solution and frozen at –80 °C for further immunoblot analysis. The resuspended proteins were analyzed by immunoblotting with anti-GFP (Santa Cruz Biotechnology Cat. No. D0314) and anti-Actin antibodies (Sigma-Aldrich Cat No. A3853). Total proteins extracted from the mycelia were also included in the analysis as a control.

**Yeast two-hybrid screening and confirmation**. The yeast two-hybrid assay[53] was used to screen cDNA library of *Arabidopsis thaliana* to identify potential targets of SsPINE1 and to further confirm the specific interaction between SsPINE1 and AtPGIP1. *SsPINE1* without signal peptide (*SsPINE1*[ΔSP]) was amplified from cDNA and used to construct the bait plasmid pGBKT7-E1. The bait strain (Y2HGold/pGBKT7-E1) was mated (in about 20 to 1 ratio) with the UY187 yeast strain (harboring the normalized Mate & Plate[TM] Libraries-Universal Arabidopsis), and the matings were selected on the SD/-Leu/-Trp (DDO) agar plates (200 ul per 150 mm plate), along with positive and negative controls following manufacturer's instructions. The emerging colonies on the DDO plates were subjected to three rounds of higher stringency selection on SD/- Ade/-His/-Leu/-Trp/-X-α-Gal/Aureobasidin A (QDO/X/A) agar plates to obtain pure blue colonies. The inserts in the prey vector were confirmed using yeast colony PCR and directly sequenced using T7 primers. The insert sequences (potential interactors) were subjected to BLASTN and BLASTX (http://www.ncbi.nlm.nih.gov/) analyses for identification and confirming the correct orientation of the insert sequences and to rule out any false-positive or large ORFs in wrong reading frames. Prey plasmids were then rescued from the candidate colonies and co-transformed with the empty vector pGBKT7 or bait vector pGBKT7-E1, separately, into Y2HGold competent yeast cells. All transformations were plated on QDO/X/A agar plates, along with positive and negative controls provided by the manufacturer.

To further confirm the specific interaction, yeast two-hybrid assay between SsPINE1 and AtPGIP1 was carried out reciprocally e. g. SsPINE1 was used as the bait and AtPGIP1 as the prey and vice versa. Y2H analysis was performed using a GAL4- based Y2H system (Matchmaker Gold Systems; Clontech, Palo Alto, CA, USA). The cDNA coding regions of AtPGIP1 and SsPINE1 were amplified by PCR with primers containing restriction sites (Supplementary Table 4), and the amplified fragments were inserted into pGADT7 and pGBKT7, respectively. The resulting bait and prey vectors confirmed by sequencing were co-transformed in pairs into the yeast strain Gold (Clontech). Transformants were grown on high stringency selective medium (SD/ -His, -Ade, -Leu, -Trp) containing aureobasidin A (AbA) and assayed by X-α-gal staining.

**Bimolecular fluorescence complementation**. Bimolecular fluorescence complementation (BiFC) analysis[33,54], was used to confirm interactions between SsPINE1 and AtPGIP1 and between SsPG1 and AtPGIP1 in vivo, with the following modifications. *SsPINE1, SsPINE1*[ΔSP] and *SsPG1* cDNAs without stop codon were amplified with appropriate primers pairs (Supplementary Table 2) and cloned into the SalI/SmaI sites of and fused with the C terminal half of red fluorescent protein in the plasmid pCNRCM3[33], resulting in plasmids pCNRCM-SsPINE1, pCNRCM-SsPINE1[ΔSP] and pCNRCM-SsPG1, respectively. The *AtPGIP1* cDNA without stop codon was amplified with primer pair AtPGIP1-Sal1-L/AtPGIP1-Sam1-R (Supplementary Table 2) and cloned into the SalI/SmaI sites of and fused with the N terminus of RFP in the vector pCNRNF3[33], resulting plasmid pCNRNF-AtPGIP1. *Agrobacterium tumefaciens* strain carrying the plasmid pCNRNF-AtPGIP1 was mixed with *A. tumefaciens* strains carrying appropriate plasmids pCNRCM-SsPINE1, pCNRCM-SsPINE1[ΔSP] or pCNRCM-SsPG1 and co-agroinfiltrated into 4-wk old leaves of *Nicotiana benthamiana* plants grown at 20 °C under long day (16 h:8 h, light:dark) conditions. Fluorescence was monitored under a confocal laser scanning microscope (Leica DM IRE2) 48 h after agroinfiltration.

**Co-immunoprecipitation (Co-IP) assays and immunoblotting analyses**. *SsPINE1* and *SsPG1* without stop codons were amplified from cDNA library using

appropriate primers (SsPINE1-Sal1-L/ SsPINE1-Sam1-R for SsPINE1; SsPG1-Sal1-L/SsPG1-Sam1-R for SsPG1, Supplementary Table 4), cloned into the *Sal* I and *Sam* I sites and fused with GFP in pCNG, resulting in the plasmids pCNG-SsPINE1 and pCNG-SsPG1, respectively (Supplementary Table 3). AtPGIP1 without stop codon was amplified from cDNA by PCR using the primer pair AtPGIP1-Sal1-L/ AtPGIP1-Sam1-R (Supplemental Information Table 4), and cloned into pCNF3 (containing 3xFlag and 35S promoter)[33], resulting in the plasmid pCHNF3-AtPGIP1 (Supplementary Table 3). *A. tumefaciens* strains containing the pCNG-SsPINE1 and pCHNF3-AtPGIP1 constructs were co-infiltrated in *N. benthamiana*. Likewise, *A. tumefaciens* strains containing the pCNG-SsPG1 and pCHNF3-AtPGIP1 constructs were used in co-infiltration. Coimmunoprecipitation was performed with leaf lysate from the co-infiltrated leaves of *N. benthamiana* 48 h after ago-infiltration. Immunoblots were performed as previously described[33].

**Yeast expression and purification of SsPINE1, SsPG1 and AtPGIP1**. The yeast expression system (Invitrogen, Carlsbad, CA) using the methylotrophic yeast *Pichia pastoris* GS115 was used to express AtPGIP1 and the fusion proteins of SsPINE1-GFP and SsPG1-GFP. The yeast expression plasmid pPIC9K (Invitrogen) was used to express the targeted genes in *P. pastoris* GS115, grown in yeast extract-peptone-dextrose (YPD) medium at 30 °C. The fusion gene SsPINE1-GFP without signal peptide was amplified with primers YEST-E1ECOR1-L and YESTE1GFP-NOT-R) from the CO-IP plasmid pCNG-SsPINE1 and cloned into the *Eco*RI and *Not*I sites of the plasmid pPIC9K, resulting in the expression vector pPIC-SsPINE1 (Supplementary Table 3). Similarly, SsPG1-GFP without signal peptide was amplified using primers YEST-PG1ECOR1-L/ YEST-PG1NOT-R from pCNG-SsPG1, and AtPGIP1 without signal peptide using primers YEST-AD1RCOR1-L/ YEST-AD1NOT1-R from pCHNF3-AtPGIP1 and cloned into plasmid pPIK9K, resulting in plasmids pPIC-SsPG1 and pPIC-AtPGIP1, respectively (Supplementary Table 3). The expression vectors were linearized with endonuclease SacI and used to transform the yeast *P. pastorisi* GS115 by using Frozen-EZ Yeast Transformation II kit (Zymo Research, Irvine, CA 92614, U.S.A).

The yeast transformants were cultured on yeast nitrogen base (YNB) agar plates without histidine. Transformants were grown on media containing different concentrations of Geneticin (G418) (Invitrogen) ranging from 0.25 to 4 mg/ml. The recombinant strains with a phenotype of *Mut*[+] were first grown in buffered minimal glycerol (BMG) medium (1.34% YNB, $4 \times 10^{-5}$% biotin, and 1% glycerol) to a final $OD_{600}$ ranging from 2 to 6, and then harvested by centrifugation. The cell pellet was suspended in buffered methanol-complex medium (BMMY), which contains 1% yeast extract, 2% peptone, 100 mM potassium phosphate with pH of 6.0, 1.34% YNB and $4 \times 10^{-5}$% biotin, to $OD_{600}$ of 1.0. The initial methanol content in BMMY was 0.5%. To maintain induction, methanol was added to a concentration of 0.5% every 24 h. The cells were incubated at 30 °C. The expressed proteins in the supernatants of the cultures were purified using affinity chromatography using Ni-NTA Superflow resin (Qiagen) in phosphate buffered saline (PBS) buffer. The purified proteins were checked by SDS-PAGE and their concentrations were estimated using Nanodrop (ND-100 Spectrophotometer, Wilmington, DE, USA) at A280. The purified proteins were analyzed by immunoblotting with anti-6X His EPITOPE TAG (RABBIT) Antibody (ROCKLAND Cat No.666-401-382S)

**Enzyme activity and enzyme inhibition assays**. PG activity was measured using two methods: the modified agarose diffusion assay[55] and the improved dinitrosalicylic acid (DNSA) method measuring the released D-galacturonic acid by PG activity[56]. Wells (5 mm dia.) were made on agar plates containing 0.5% poly-galacturonic acid and 0.8% agarose in 100 mM sodium acetate buffer (pH 4.6). Test proteins or a combination of the proteins in a total volume of 100 μl were added to the wells and the assay plates were incubated for 12 h at 30 °C. Then, the plates were washed with 6 N HCl to precipitate polygalacturonic acid and forming a halo ring between soluble partially digested PGA because of enzyme activity. The diameter of the halo ring was measured using a vernier caliper. The purified SsPG1, SsPINE1 and AtPGIP1 proteins were first tested separately for any PG activity. Since only SsPG1 possessed PG activity. SsPG1 was tested in various concentrations to show quantitative nature of the enzyme assay. In the DNSA method[56], AtPGIP1 (10 μg) or AtPGIP1 and SsPINE1 (10 μg ea.) were added to 0.5 μg SsPG1 in 225-μl volume (0.5% polygalacturonic acid in 100 mM sodium acetate buffer). After incubation at 30 °C for up to 6 h, 750 μl DNSA solution was added to stop the reaction. The mixture was then incubated at 100 °C on a water bath for 5 min. 100-μl aliquot was used for measuring absorbance at 575 nm against the blank control using a SpectraMax® M5microplate reader. The calibration was carried out using D-galacturonic acid as the standard.

**Relative affinity of SsPINE1 and SsPG1 with AtPGIP1**. Initially, a modified BiFC method was used to verify that the three proteins SsPINE1, SsPG1 and AtPGIP1 were co-located *in planta*. The two BiFC plasmid AtPGIP1-nRFP and SsPINE1-cRFP together with SsPG1-GFP were co-expressed in tobacco leaves and the locations of the fluorescence signals of GFP and the complemented RFP were monitored 2 days after agroinfiltration under a confocal laser scanning microscope. Then, two methods were employed to investigate the relative affinity of SsPINE1

and SsPG1 with AtPGIP1. First, in a modified Co-IP experiment, we used three constructs *SsPINE1-GFP*, *SsPG1-GFP* and *AtPGIP1-3xFlag* to co-infiltrate tobacco leaves, in addition to conventional Co-IPs described above. The relative intensity of SsPINE1 and SsPG1 that co-precipitated with AtPGIP1 was detected in immunoblot analysis. As negative controls, 35 S::3xFlag was used instead of AtPGIP1-3xFlag.

The second method was another modified Co-IP assay. We tested whether addition of SsPINE1 could dissociate SsPG1 from AtPGIP1 since the PGIP-PG binding is reversable[39]. Agro-coinfiltration of tobacco leaves was performed as described for the conventional Co-IP. Before performing immuno-precipitation, the competing protein obtained from yeast expression was added in increasing concentrations to the leaf lysate. Specifically, the SsPINE1-GFP protein (0, 100 ng and 10 µg) was added to equal volume of the same leaf lysate from leaves co-infiltrated with SsPG1-GFP and AtPGIP1-3xFlag constructs and incubated for 30 min, prior to immunoprecipitation with anti-Flag antibody. Likewise, the purified SsPG1-GFP protein (0, 100 ng and 10 µg) was added to equal volume of the same leaf lysate from leaves co-infiltrated with the SsPINE1-GFP and AtPGIP1-3xFlag constructs and incubated for 30 min. After the incubation, co-immunoprecipitation and immuno-blotting assays were performed as described above for conventional Co-IP.

**Site-directed mutagenesis of AtPGIP1**. AtPGIP1 has the typical extracytoplasmic type with ten plant-specific leucine-rich repeat (LRR) modules[35]. Residues involved in the interaction with PGs are located in the concave surface formed by the ß-sheet. The concave surface of AtPGIP1 was examined by conducting point mutagenesis on three regions of the β-sheet, creating three mutation variant P1, P2 and P3. P1 involved amino acid substitutions in LRR1 to LRR3, P2 in LRR4 to LRR7, and P3 LRR8 to LRR10. The specific amino acid substitutions are shown in Supplementary Fig. 7a. The secondary folding structures of the mutation variants as well as the wildtype AtPGIP1 were checked at the Swiss-Model website (https://www.swissmodel.expasy.org/) to ensure that the amino acid substitutions did not disrupt the folding structure. The nucleotide sequences of the variants P1, P2 and P3 were synthesized commercially at the Gene Universal (Newark, DE, USA), and the synthesized sequences were confirmed by sequencing at the Washington State University sequencing core facility before used in cloning. AtPGIP1, P1, P2 and P3 were cloned into plasmid pPIK9K that includes the poly-histidine tag. The expression vectors were linearized with endonuclease SacI and used to transform the yeast *P. pastorisi* GS115 using Frozen-EZ Yeast Transformation II kit (Zymo Research, Irvine, CA 92614, U.S.A). Procedures of yeast transformation and culturing, protein purification and Western blot detection and enzyme inhibition assays are as described above.

**Transformation of *Arabidopsis thaliana* with SsPINE1 and AtPGIP1 and pathogenicity assays**. The plasmids pCNG-SsPINE1 and pCHNF3-AtPGIP1 used in Co-IP were used to transform *A. thaliana* Col-0 plants, using the floral dip method with kanamycin as a selection marker[57]. T1 seeds were selected on ½ MS medium containing kanamycin (50 µg /ml). T3 seeds were used for all experiments. Expression of SsPINE1-GFP in the transgenic plants was verified by Western blot with anti-GFP antibody and fluorescence microscopy. Overexpression of AtPGIP1-3xFlag in the transgenic plants was verified by Western blot with anti-Flag antibody and RT-qPCR and normalized using *UBQ5* gene expression. Secretion of AtPGIP1 was tested in the AtPGIP1-overexpressing line using Western blot of the apoplastic fluid which was collected by centrifugation of leaf tissue. The growth rate and morphology of the transgenic lines were compared with the wildtype Col-0 plants.

*Arabidopsis* plants were grown at 20 °C in a growth chamber under long day (16 h:8 h, light:dark) conditions. The reactions of 4-week old plants of the wildtype (Col-0) and *SsPINE1*-expressing and *AtPGIP1*-overexpressing lines to Sclerotinia infection were assessed using the following pathogenicity assay. Mycelia plugs (4 mm diam.) from the margins of actively growing 2-day old colonies of *S. sclerotiorum* strains wild type (WMA1), Δ*SsPINE1* and Δ*SsPG1* mutants on SY medium were used for inoculation. Virulence was quantified using two methods: macroscopically measuring lesion size with a caliper and molecularly measuring the ratios of *S. sclerotiorum* DNA to *Arabidopsis thaliana* DNA in infected leaves at 48 hpi via genomic DNA quantitative PCR, for which the *S. sclerotiorum Actin* gene and *Arabidopsis AtUBQ5* gene (primers in Supplementary Table 2) were amplified. For determining relative biomass, due to the small tissue mass two of the six replicate leaves were randomly combined to represent one replicate for DNA isolation. Each treatment was carried out with 6 (lesion size) or 3 (biomass) replicates. The experiment was performed three times.

**RNA extraction and quantitative real-time PCR**. Total RNA samples of fungi and plants were isolated using RNeasy Mini Kit (QIAGEN) according to the manufacturer's protocols. Contaminating genomic DNA was eliminated by RNase-free Recombinant DNase I (Thermo Fisher Scientific). The first-strand cDNAs were synthesized by M-MLV Reverse Transcriptase (Applied Biosystems), and quantitative real-time reverse transcription PCRs (RT-qPCRs) were carried out in a CFX96 Real-Time PCR Detection System (Bio-Rad) with iTaq universal SYBR Green supermix (Bio-Rad). The *S. sclerotiorum* β-tubulin gene *Sstub1*

(SS1G_04652) and *A. thaliana* UBQ5 (AT3G62250) were used to normalize the RNA samples for each real-time RT-PCR. For each gene, real-time RT-PCR assays were repeated at least twice, each with three technical replicates.

**Genetic manipulation in *Botrytis cinerea***. The SsPINE1 protein sequence was used as query in protein BLAST search of the NCBI protein database. Representative accessions from the BLAST search results showing high similarities were selected for phylogenetic analysis. A closely related homolog in *Botrytis cinerea* was selected for further investigation. BcPINE1 (*BC1G_04506*) in *Botrytis cinerea* strain B05.10 was replaced with the hygromycin resistance cassette (*hph*)[58]. The gene replacement construct was generated as described[58]. The 5′ (536 bp) and 3′ (513 bp) flanking regions of the *BcPINE1* open reading frame were amplified by PCR with specific primers (Supplementary Table 2) from genomic DNA of the wild-type strain B05.10, and the PCR fragments were cloned into the upstream and downstream regions, respectively, of the *hph* cassette using the Gibson Assembly MasterMix kit (New England Biolabs). The transformants generated by protoplast transformation were selected on PDA containing 70 µg/ml hygromycin and further purified through single conidia. Homokaryotic BcPINE1 replacement mutants were verified using PCR with primers BoE15-L/BoE13-R (Supplementary Table 2) specific for the BcPINE1 (*BC1G_04506*) locus.

Virulence of the *B. cinerea* isolates was assess by inoculating on *A. thaliana* leaves as described by Schumacher et al.,[59]. The leaves of 4-week-old *A. thaliana* plants were inoculated with conidial suspensions (10 µl, $2 \times 10^5$ conidia/ml) and incubated in a humid chamber at 22 °C for 48 h. Virulence was quantified by measuring lesion size and quantifying the ratios of *B. cinerea* DNA to host plant DNA at 48 hpi via genomic DNA quantitative PCR, for which the *B. cinerea Bcgpdh* gene (primers in Supplementary Table 2) and *Arabidopsis AtUBQ5* gene were amplified. For relative biomass determination, two of the six replicate leaves were randomly combined to form one replicate for DNA isolation. Each treatment was carried out with 6 (lesion size) or 3 (biomass) replicates and each experiment was performed three times.

**Reporting summary**. Further information on research design is available in the Nature Research Reporting Summary linked to this article.

## Data availability

The source data underlying Figs. 1–7 and Supplementary Figs. 1, 3, 4, 5, 8 and 9 are provided as a Source data file.

The accession numbers for the *Sclerotinia sclerotiorum* proteins, Arabidopsis thaliana proteins and SsPINE1 homologs analyzed in this manuscript are: *Sclerotinia sclerotiorum* SsPINE1 [APA12722.1], SsPG1 [AAM22186.1], SsPG3 [AY312510.2], SsPG5 [XM_001594320.1], SsPG6 [AF501308.1], Arabidopsis thaliana AtPGIP1 [NP_196304.1], AtPGIP2 [NP_196305.1], At2G35790 [NP_565823.1], At2G28100 [NP_180377.2], AtNdhL [XP_002887327.1], *Botrytis cinerea* BcPINE1 [XP_001557256.1]. Source data are provided with this paper.

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

## Acknowledgements

The research was supported in part by the USDA National Sclerotinia Initiative to W.C. and by the National Natural Science Foundation of China grant number 31701737 to W.W.

## Author contributions

W.W. conducted gene replacement, in vivo experiments, western blotting, BiFC and Co-IP assays, protein expression in yeast and enzymatic assays, *Arabidopsis* transformation, RNA-seq and disease assays; L.X. identified SsPINE1 and its target AtPGIP1; H.P. conducted genomic analysis and writing; W.Z. performed gene knockout in *Botrytis cinerea*; K.T. J.C. contributed to experimental design and writing; K.S., G.V. provided relevant materials and resources; W.C., W.W. and H.P. wrote the manuscript. All authors contributed to the editing of the manuscript.

## Competing interests

The authors declare no competing interests.
