## [Peer Review File · Nature Communications]

A novel fungal extracellular effector inactivates plant polygalacturonase-inhibiting proteinREVIEWER COMMENTS

Reviewer #1 (Remarks to the Author):

The authors describe the identification of an apoplastic effector protein of the broad host range plant pathogenic fungus, *Sclerotinia sclerotiorum*, called SsPINE1, that targets PGIPs, thereby suppressing plant defence against fungal polygalacturonases (SsPG1 and others). Their claim is based on interaction studies (using yeast-2-hybrid, BiFC and Co-IP) of SsPINE1 and AtPGIP1, SsPG1 and AtPGIP1, reduced virulence of SsPINE1 and SsPG1 mutants, competition experiments between SsPINE1 and SsPG1 for their binding to AtPGIP1, and expression of SsPINE1 which leads to increased sensitivity to *B. cinerea* infection.

This is a very interesting study which reveals a new mode of action of an apoplastic fungal effector, which plays a potentially important role for pathogenesis not only in *Sclerotinia* but also other necrotrophic plant pathogens. Therefore I think that it is principally worth to be published in this journal.

However, I observed several points that need to be addressed and clarified before:

- Size markers should be shown in the Western blots, otherwise it is very tedious to sort out which band might correspond to which protein.
- The infection experiments with the PINE1 mutants have been made on pea, which is admittedly a more natural host for *Sclerotinia* than *Arabidopsis*. Nevertheless, since the molecular studies were performed with AtPGIP1, the authors should also present the effects of PINE1 knockout on *Arabidopsis* infection.
- Fig. 3a: Expression of SsPGINE1-cRFP/ AtPGIP1-nRFP is not uniform, and the pictures show nothing else than an apoplastic location of the three proteins, which has been shown before. I would delete Fig. 3a.
- In Fig. 4, the authors show that purified AtPGIP (which conc.?) suppresses fungal infection. Because they generated AtPGIP mutants with low affinity to SsPG1 (Suppl Fig. 7), one of these mutant proteins should be used as control to demonstrate that the inhibitory activity of AtPGIP is dependent on its affinity to PG1.
- For detection of PG activity, the authors use an agar diffusion assay which has only a small dynamic range (see Fig. 5C, supp. Fig. 6). I have difficulties to comprehend how this assay is able to produce diagrams as in Fig. 4a and 4b. Because PG1 inhibition by AtPGIP and release from this inhibition are crucial for this work, the authors should use regular enzyme assays for polygalacturonase (which are even commercially available at Sigma etc.).
- In Suppl. Fig. 9, a repeat structure of SsPINE1 is indicated, this needs to be made clear by means of an alignment between the two internal repeats. Furthermore, the alignment in b does not make clear which aa are conserved between the fungi; this should be improved.

- To my opinion, Fig. 8 does not contribute much to the understanding of findings in the paper. Furthermore, the dots indicating ,plant signal' and ,unknown resistance signal' are not related to this work and confusing by their oversimplification.
- The authors write in the discussion (l. 376ff.): ,BcPINE1, a homolog of SsPINE1 in *B. cinerea*, also interacts with AtPGIP1 and contributes to virulence, suggesting that the SsPINE1-mediated virulence mechanism has evolved across a range of plant pathogens under the selection pressure exerted by' host PGIPs.' This is not appropriate: PINE1 is restricted to a small range of mostly necrotrophic genera of the order Helotiales (*Sclerotinia*, *Botrytis*, *Monilinia*).
- A general comment: It has become clear that PGIPs have a specificity for certain fungal PGs, and are unable to inhibit all of them. In this manuscript, it appears AtPGIP1 is highly active against SsPG1, and that SsPINE1 has an even higher affinity to AtPGIP1 than SsPG1. The effects of PINE1 have been tested against AtPGIP1, but mutant analysis has shown that the effector is also active against major soybean PGIPs. Of course, it would be interesting to test the affinity of PINE1 against different PGIPs, but this would be beyond the scope of this work. In any case, the authors should avoid to simplify to complexity of the PGIP-PG-PINE1 interaction.

Reviewer #2 (Remarks to the Author):

The article by Wei et al describes a novel effector (SsPINE1) of the necrotrophic fungus *Sclerotinia sclerotiorum*, that inactivate the PG-inhibiting protein 1 (PGIP1) from *Arabidopsis*, that is a protein inhibitor that counteract the activity of PG, a fungal cell wall degrading enzyme. *Arabidopsis* PGIP1 function is the protection of cell wall and the mitigation of its hydrolysis by fungal PGs during fungal colonization. This novel effector protein (PGIP-Inactivating Effector 1, or SsPINE1) is the first one described to counteract the activity of plants PGIPs and therefore an example of an additional layer of complexity in plant-pathogen interaction. The article nicely shows that ectopic expression of SsPINE1 in plant reduces its resistance against *S. sclerotiorum*. Also, using co-IP experiments it is showed that PINE1 directly interact with PGIP1 and displaces PGs binding by PGIP1. Since PINE1 seems to be conserved in other fungal pathogens like *Botrytis cinerea*, it can be anticipated that PINE effectors play important roles in necrotrophic virulence through inactivating plant PGIP. The article contains novel and valuable information, but some additional experiments (particularly controls) should be included to further demonstrate and support some of the key conclusions.

Major Comments:

1. Demonstration of SsPINE1 interaction with AtPGIP1 is supported by various data including physically interaction in three systems: yeast two-hybrid, BiFC and Co-IP. However, in some of these experiments negative controls are lacking and this might be relevant since the author indicate that in the screening in

Arabidopsis 108 prey (proteins) were found to interact with SsPINE1. This opens the question about the specificity of SsPINE1 binding to AtPGIP1, and whether StPINE1 also binds to other PGIP of Arabidopsis (five members). This specificity of binding must be proved, the list of prey proteins identified should be listed, and some additional control (negative) of SsPINE1 interaction with protein of the list should be included for comparison.

2. The article clearly shows that deletion of SsPINE1 reduces *S. sclerotiorum* virulence. In contrast, ectopic expression of SsPINE1 in plant reduces plant resistance against *S. sclerotiorum*. However, in some of the results generated with the transgenic plants of Arabidopsis some data of virulence do not match with the expected results and hypothesis. The authors indicate that this might be due to the disease resistance index is not enough accurate. For this and other reasons the estimation of disease index in inoculated plants should be performed by quantifying the fungal biomass in inoculated plants (standard in the field) and not just base on macroscopic necrotic lesions determination.

3. BiFC and Co-IP results demonstrate physical interaction between SsPG1 and AtPGIP1 and PG-PGIP (this last consistent with previous data). The authors performed competition assays using CoIPs with three constructs SsPINE1-GFP, SsPG1-GFP and AtPGIP1-3xFlag that were co-infiltrated into tobacco leaves. Flag IP detected significantly more SsPINE1-GFP than SsPG1-GFP and they suggest that these data indicate that SsPINE1 has higher binding affinity than SsPG1 to AtPGIP1. Additional assays, like ITC or thermophoresis, would be required to further prove these data.

4. The results suggest that both P2 and P3 regions are involved in AtPGIP1-mediated inhibition of SsPG1, with the P2 region playing a more critical role. The data indicate that variants P2 and P3 of AtPGIP1 exhibited approximately 5% and 40% inhibitory activities, respectively. These data are interesting, but lack of additional controls like for example P2 and P3 regions of other AtPGIPs. There is not any indication in the article about the other 5 PGIPs of Arabidopsis (were among the preys?). In general experiments to validate the specificity of SsPINE1 for AtPGIP1 are lacking. For example, SsPG1-deletion mutant did cause significant necrotic lesions on SsPINE1-expressing plants, and this clearly indicate that SsPINE1 is mitigating PGIP activity (only AtPGIP1?) on other SsPGs necrotrophic activities. The conclusion of the authors is that "overexpression of SsPINE1 in the host plant aided Sclerotinia virulence factors other than SsPG1, likely other SsPGs". For this reason, it is necessary to determine the specificity and interactions of SsPINE1 (see comment 3).

5. The article shows that overexpression of AtPGIP1 enhanced plant immunity to the Δ SsPINE1 mutant, but such enhancement is not obvious in the wild-type strain or the Δ SsPG1 mutant (Fig. 6). The explanation of the authors for this contradictory results is that "consistent with previous reports that the high susceptibility of *A. thaliana* makes it difficult to differentiate genotypic differences in response to *S. sclerotiorum* infection". This explanation can not be accepted for an high quality journal and it is a clear indication that the authors should develop an additional method for the determination of fungal infection progression in inoculated plants that should be based on fungal biomass quantification, that is an standard methodology in the field. Infection based on leave necrosis cause by fungal infection might lead to misinterpretation of some results, particularly in the case of necrotrophic fungi that secrete compounds like oxalic acid and other effectors that induce plant cell necrosis.

Minor points:

1. Figure S1 a: the bands of the gels are not in the same orientation (SsE1 vs Actin). Are these bands from the same experiment? This applies to several figures in the article.
2. Figure S1D: Lack the control of mycelium.
3. Line 261: "All three mutation variants of AtPGIP1 were shown to have folding structure similar to the native AtPGIP1 (not shown)". These data MUST be shown. I understand that these are structural predictions.
4. Line 363: "Yet, *S. Sclerotiorum*", probably Y et. Must be deleted
5. Some recent papers describe the implication of PGIPs (e.g. OsPGIP1) in regulating other virulence factors in addition to PGs. See also comments 3 and 4 about specificity.
6. Figure 2C. Western of the panel input is not coincident with upper panel, why? Similar discrepancies are detected in other Western experiments (Figure 5 and 6). How many repetitions were performed for these experiments? This should be indicated in the legend to figure.
7. Supplementary Figure 2: Why Anti-Myc antibody is used in these experiments?

Point-to-Point Response to Reviews
(Responses in Blue)

We want to express our appreciations to the two reviewers for their constructive criticism and valuable comments and suggestions to improve this manuscript. We made every effort to follow the suggestions as explained in the following point-by-point response. As a result, we believe the manuscript has been improved substantially. Thanks again to both reviewers.

The authors describe the identification of an apoplastic effector protein of the broad host range plant pathogenic fungus, *Sclerotinia sclerotiorum*, called SsPINE1, that targets PGIPs, thereby suppressing plant defence against fungal polygalacturonases (SsPG1 and others). Their claim is based on interaction studies (using yeast-2-hybrid, BiFC and Co-IP) of SsPINE1 and AtPGIP1, SsPG1 and AtPGIP1, reduced virulence of SsPINE1 and SsPG1 mutants, competition experiments between SsPINE1 and SsPG1 for their binding to AtPGIP1, and expression of SsPINE1 which leads to increased sensitivity to *B. cinerea* infection.

This is a very interesting study which reveals a new mode of action of an apoplastic fungal effector, which plays a potentially important role for pathogenesis not only in *Sclerotinia* but also other necrotrophic plant pathogens. Therefore I think that it is principally worth to be published in this journal.

However, I observed several points that need to be addressed and clarified before:

- Size markers should be shown in the Western blots, otherwise it is very tedious to sort out which band might correspond to which protein.

> As suggested, we added marker sizes to blot figures wherever applicable.

- The infection experiments with the PINE1 mutants have been made on pea, which is admittedly a more natural host for *Sclerotinia* than *Arabidopsis*. Nevertheless, since the molecular studies were performed with AtPGIP1, the authors should also present the effects of PINE1 knockout on *Arabidopsis* infection.

> We agree! Considering that the effect of PINE1 knockout mutants on infection of wild type *Arabidopsis* are shown in Fig. 6 along with transgenic *Arabidopsis*, we don't feel the need to duplicate a portion of Figure 6. Hope this is acceptable.

- Fig. 3a: Expression of SsPGINE1-cRFP/ AtPGIP1-nRFP is not uniform, and the pictures show nothing else than an apoplastic location of the three proteins, which has been shown before. I would delete Fig. 3a.

> The original Fig. 3a was deleted from the figure as suggested. The figure is now shown as a supplementary figure in Supplementary Fig. 2c, as we feel it is important to show that the three proteins all co-localized together, as a base for conducting the binding competition experiments.

- In Fig. 4, the authors show that purified AtPGIP (which conc.?) suppresses fungal infection. Because they generated AtPGIP mutants with low affinity to SsPG1 (Suppl Fig. 7), one of these mutant proteins should be used as control to demonstrate that the inhibitory activity of AtPGIP is dependent on its affinity to PG1.

> As suggested, the concentrations of the experimental proteins are indicated where appropriate. We included the variant protein PGIP1-P2 in bioassays on leaf infection (Fig 4c, d).

- For detection of PG activity, the authors use an agar diffusion assay which has only a small dynamic range (see Fig. 5C, supp. Fig. 6). I have difficulties to comprehend how this assay is able to produce

diagrams as in Fig. 4a and 4b. Because PG1 inhibition by AtPGIP and release from this inhibition are crucial for this work, the authors should use regular enzyme assays for polygalacturonase (which are even commercially available at Sigma etc.).

> As suggested, we used the colorimetric dinatrosalicylic acid (DNSA) method (Wang et al., 1997, Ref #56) in measuring the released D-galacturonic acid due to PG activity and updated the results. The sensitivity of the DNSA method helped us capture the dynamics of the interactions among the three proteins (Fig. 4b).

- In Suppl. Fig. 9, a repeat structure of SsPINE1 is indicated, this needs to be made clear by means of an alignment between the two internal repeats. Furthermore, the alignment in b does not make clear which aa are conserved between the fungi; this should be improved.

>As suggested, the two internal repeat sequences are aligned and the conserved amino acid residues among the PINE1 sequences are indicated with different shades of blue in the multiple sequence alignment (Supplementary Fig. 10).

- To my opinion, Fig. 8 does not contribute much to the understanding of findings in the paper. Furthermore, the dots indicating 'plant signal' and 'unknown resistance signal' are not related to this work and confusing by their oversimplification.

> Fig. 8 is deleted as suggested.

- The authors write in the discussion (l. 376ff.): 'BcPINE1, a homolog of SsPINE1 in *B. cinerea*, also interacts with AtPGIP1 and contributes to virulence, suggesting that the SsPINE1-mediated virulence mechanism has evolved across a range of plant pathogens under the selection pressure exerted by host PGIPs.' This is not appropriate: PINE1 is restricted to a small range of mostly necrotrophic genera of the order Helotiales (*Sclerotinia*, *Botrytis*, *Monilinia*).

> We agree. Although PINE1 exists in a large number of fungal species, the list is limited to species within the order Helotiales. We modified this statement as "BcPINE1, a homolog of SsPINE1 in *B. cinerea*, also interacts with AtPGIP1 and contributes to virulence, suggesting that SsPINE1-mediated virulence mechanism is conserved among broad host range necrotrophic fungi in the order *Helotiales* under the selection pressure exerted by host PGIPs." (Lines 400-403).

- A general comment: It has become clear that PGIPs have a specificity for certain fungal PGs, and are unable to inhibit all of them. In this manuscript, it appears AtPGIP1 is highly active against SsPG1, and that SsPINE1 has an even higher affinity to AtPGIP1 than SsPG1. The effects of PINE1 have been tested against AtPGIP1, but mutant analysis has shown that the effector is also active against major soybean PGIPs. Of course, it would be interesting to test the affinity of PINE1 against different PGIPs, but this would be beyond the scope of this work. In any case, the authors should avoid to simplify to complexity of the PGIP-PG-PINE1 interaction.

> There are two PGIP genes in *Arabidopsis thaliana* (Ref #35). Following the suggestion, we tested and found that SsPINE1 does not interact with AtPGIP2 in yeast two-hybrid assays (Supplementary Fig. 2a).

As the reviewer advised, we have added the text on lines 405-408 to avoid simplifying the complexity of the tripartite PGIP-PG-PINE interaction.

Reviewer #2 (Remarks to the Author):

The article by Wei et al describes a novel effector (SsPINE1) of the necrotrophic fungus *Sclerotinia sclerotiorum*, that inactivate the PG-inhibiting protein 1 (PGIP1) from *Arabidopsis*, that is a protein inhibitor that counteract the activity of PG, a fungal cell wall degrading enzyme. *Arabidopsis* PGIP1 function is the protection of cell wall and the mitigation of its hydrolysis by fungal PGs during fungal colonization. This novel effector protein (PGIP-Inactivating Effector 1, or SsPINE1) is the first one described to counteract the activity of plants PGIPs and therefore an example of an additional layer of complexity in plant-pathogen interaction. The article nicely shows that ectopic expression of SsPINE1 in plant reduces its resistance against *S. sclerotiorum*. Also, using co-IP experiments it is showed that PINE1 directly interact with PGIP1 and displaces PGs binding by PGIP1. Since PINE1 seems to be conserved in other fungal pathogens like *Botrytis cinerea*, it can be anticipated that PINE effectors play important roles in necrotrophic virulence through inactivating plant PGIP. The article contains novel and valuable information, but some additional experiments (particularly controls) should be included to further demonstrate and support some of the key conclusions.

Major Comments:

1. Demonstration of SsPINE1 interaction with AtPGIP1 is supported by various data including physically interaction in three systems: yeast two-hybrid, BiFC and Co-IP. However, in some of these experiments negative controls are lacking and this might be relevant since the author indicate that in the screening in *Arabidopsis* 108 prey (proteins) were found to interact with SsPINE1. This opens the question about the specificity of SsPINE1 binding to AtPGIP1, and whether StPINE1 also binds to other PGIP of *Arabidopsis* (five members). This specificity of binding must be proved, the list of prey proteins identified should be listed, and some additional control (negative) of SsPINE1 interaction with protein of the list should be included for comparison.

> Following the suggestion from the reviewer, we have included more controls in the experiments. We tested potential interactions of SsPINE1 with the other member (AtPGIP2) of the *Arabidopsis* PGIP gene family and found that SsPINE1 does not interact with AtPGIP2 in yeast 2-hybrid assays in either combination with SsPINE1 as the bait or as the prey (Supplementary Fig 2a). There are only two PGIP genes in *Arabidopsis thaliana* (Ferrari et al. 2002, reference #35). We also tested three other prey proteins of *A. thaliana* that showed up in both Y2H screens (Supplementary Table 1). Y2H assays showed that SsPINE1 did not interact with At2G28100 and AtNdhL (NM_105744.4), but it did interact with At2G35790 (Supplementary Fig. 2b). At2G35790 is a transmembrane protein in the mitochondria and its functions in *Arabidopsis* and functions of its homologues in other plants are unknown. We believe that the interaction between SsPINE1 and At2G35790 in Y2H assay is an artifact of the assay and lacks biological relevance because they are unlikely to be in contact *in planta* (SsPINE1 in the apoplast and At2G35790 in the mitochondria). Thanks for the suggestion to include the negative controls. We now have stronger evidence to state that the SsPINE1-AtPGIP1 interaction is specific.

2. The article clearly shows that deletion of SsPINE1 reduces *S. sclerotiorum* virulence. In contrast, ectopic expression of SsPINE1 in plant reduces plant resistance against *S. sclerotiorum*. However, in some of the results generated with the transgenic plants of *Arabidopsis* some data of virulence do not match with the expected results and hypothesis. The authors indicate that this might be due to the disease resistance index is not enough accurate. For this and other reasons the estimation of disease index in inoculated plants should be performed by quantifying the fungal biomass in inoculated plants (standard in the field) and not just base on macroscopic necrotic lesions determination.

> As suggested, virulence was quantified using relative pathogen biomass in addition to using traditional necrotic lesion area. Biomass quantification has indeed improved our interpretation of the response of AtPGIP1-overexpressing line to *Sclerotinia* infection (Fig. 6c).

3. BiFC and Co-IP results demonstrate physical interaction between SsPG1 and AtPGIP1 and PG-PGIP (this last consistent with previous data). The authors performed competition assays using CoIPs with three constructs SsPINE1-GFP, SsPG1-GFP and AtPGIP1-3xFlag that were co-infiltrated into tobacco leaves. Flag IP detected significantly more SsPINE1-GFP than SsPG1-GFP and they suggest that these data indicate that SsPINE1 has higher binding affinity than SsPG1 to AtPGIP1. Additional assays, like ITC or thermophoresis, would be required to further prove these data.

This has been the most challenging. The problems we face are that the proteins expressed in prokaryotic cells were not functional and that the protein yields from yeast expression were too low for ITC assay.

We first attempted to express these proteins in *E. coli*. Only SsPINE1 was successfully expressed and secreted outside the *E. coli* cells. Both SsPG1 and SsPGIP1 were expressed in the inclusion body. They lost their activities after being purified from inclusion bodies. We then switched to the yeast expression system. Induced by formaldehyde, these proteins can be expressed and secreted to the culture medium. However, their concentrations were too low. Hence, the expressed protein needs to be concentrated. We attempted to concentrate AtPGIP1 from 10 L yeast culture media, but only achieved the concentration of 70 μM that was far below the requirement for the ITC assay (>500 μM). Furthermore, AtPGIP1 activity decreased by 50% after freeze-drying. After working on SsPG1 and AtPGIP1 expression and purification for five months using different expression systems including different vectors and purification methods, we were unable to obtain proteins with sufficient quantity for the ITC assay.

In response to this criticism, we have modified texts to refrain from claiming higher affinity of SsPINE1 (Lines 218-220, 236-238, 372-373, 395-396)

4. The results suggest that both P2 and P3 regions are involved in AtPGIP1-mediated inhibition of SsPG1, with the P2 region playing a more critical role. The data indicate that variants P2 and P3 of AtPGIP1 exhibited approximately 5% and 40% inhibitory activities, respectively. These data are interesting, but lack of additional controls like for example P2 and P3 regions of other AtPGIPs. There is not any indication in the article about the other 5 PGIPs of Arabidopsis (were among the preys?). In general experiments to validate the specificity of SsPINE1 for AtPGIP1 are lacking. For example, SsPG1-deletion mutant did cause significant necrotic lesions on SsPINE1-expressing plants, and this clearly indicate that SsPINE1 is mitigating PGIP activity (only AtPGIP1?) on other SsPGs necrotrophic activities. The conclusion of the authors is that "overexpression of SsPINE1 in the host plant aided *Sclerotinia* virulence factors other than SsPG1, likely other SsPGs". For this reason, it is necessary to determine the specificity and interactions of SsPINE1 (see comment 3).

> We think this question emphasizes appropriate negative controls for specificity and overlaps with question #1. Our answer to question #1 partially addressed this question. There are only two PGIP genes in *A. thaliana* (Ref #35) and SsPINE1 interacts with only AtPGIP1 but not PGIP2 (Supplementary Fig. 2a). And AtPGIP2 was not among the 108 prey proteins in the two Y2H screens (Supplementary Table 1). AtPGIP1-P2 variant lost its binding with SsPINE1 and also with SsPG1 (Fig. 5a, b). Consistent with its lost

binding with SsPG1, AtPGIP1-P2 cannot inhibit enzymatic activity of SsPG1 (Fig. 5c, d) and *Sclerotinia* infection (Fig. 4c, d).

The necrosis caused by SsPG1-deletion mutant must be caused by other SsPGs. The fact that the SsPG1-deletion mutant caused significant necrosis on SsPINE1-expressing plant suggests that the other SsPGs were active (not inhibited) due to inactivation of AtPGIP1 by SsPINE1 expressed in the plant. Thus, AtPGIP1 inhibits other SsPGs besides SsPG1. We believe that demonstrating the specificity of SsPINE1 interacting with and inactivating AtPGIP1 is the primary importance of this study. Whether AtPGIP1 can inhibit other SsPGs in addition to SsPG1 should be a subject of future study. A broader PG-inhibition spectrum of AtPGIP1 would further support the importance of SsPINE1 in *Sclerotinia* virulence and likely virulence of other PINE1-possessing necrotrophic pathogens.

5. The article shows that overexpression of AtPGIP1 enhanced plant immunity to the Δ SsPINE1 mutant, but such enhancement is not obvious in the wild-type strain or the Δ SsPG1 mutant (Fig. 6). The explanation of the authors for this contradictory results is that “consistent with previous reports that the high susceptibility of *A. thaliana* makes it difficult to differentiate genotypic differences in response to *S. sclerotiorum* infection”. This explanation can not be accepted for an high quality journal and it is a clear indication that the authors should develop an additional method for the determination of fungal infection progression in inoculated plants that should be based on fungal biomass quantification, that is an standard methodology in the field. Infection based on leave necrosis cause by fungal infection might lead to misinterpretation of some results, particularly in the case of necrotrophic fungi that secrete compounds like oxalic acid and other effectors that induce plant cell necrosis.

> We employed the more sensitive relative pathogen biomass method. With this method we were able to show the enhanced resistance in AtPGIP1-overexpressing plant in reduction of pathogen biomass of the wildtype strains of both *S. sclerotiorum* and *B. cinerea* (Figs. 6c and 7c).

Minor points:

1. Figure S1 a: the bands of the gels are not in the same orientation (SsE1 vs Actin). Are these bands from the same experiment? This applies to several figures in the article.

> The band orientation issue is now resolved (Supplementary Fig. 1a). We attached original Western blot pictures for inspection.

2. Figure S1D: Lack the control of mycelium.

> The mycelium control is shown in secretion tests as suggested (Supplementary Figs. 1d and 5e).

3. Line 261: “All three mutation variants of AtPGIP1 were shown to have folding structure similar to the native AtPGIP1 (not shown)”. These data MUST be shown. I understand that these are structural predictions.

> The modeled 3-D structure of AtPGIP1 and its mutation variants were shown as suggested (Supplementary Fig. 7b)

4. Line 363: “Yet, *S. Sclerotiorum*”, probably Y et. Must be deleted

> Deleted as suggested.

5. Some recent papers describe the implication of PGIPs (e.g. OsPGIP1) in regulating other virulence factors in addition to PGs. See also comments 3 and 4 about specificity.

>The interaction between PGIP and PG is intricate. The involvement of PINE in this engagement makes the battle more so. Some PGIPs are shown to have functions in regulating other virulence factors in addition to inhibiting PGs. Whether the interaction of PINE with PGIP would affect the other functions of PGIP would be subjects of future investigation (lines 405-408).

6. Figure 2C. Western of the panel input is not coincident with upper panel, why? Similar discrepancies are detected in other Western experiments (Figure 5 and 6). How many repetitions were performed for Q`these experiments? This should be indicated in the legend to figure.

> We have corrected the problem in properly aligning the blots and provided original blot photos in Source data file for inspection. Information of replicates is added to the figure legends.

7. Supplementary Figure 2: Why Anti-Myc antibody is used in these experiments?

> The BiFC vectors have both FLAG and MYC tags fused with the expressed proteins (Ref #33), We indicated this fact in the figure legend to avoid confusing.

REVIEWERS' COMMENTS

Reviewer #1 (Remarks to the Author):

The authors have submitted a revised version in which they have addressed all comments of the reviewers, most of them in a fully satisfying manner and including new experimental data. They have not been able to confirm with an independent assay (ITC) because of difficulties to express SsPG1 and SsPGIP1 in sufficient amount. The difficulty of PG1 expression has been already experienced in *B. cinerea* (Kars et al., 2005, Plant J.; own unpublished results), therefore I would accept the reply of the authors.

Overall, I now recommend the manuscript for publication.

Reviewer #2 (Remarks to the Author):

In this revised version of the article, the authors have performed the majority of the experiments requested and suggested to improve the quality of experimental controls and to clarify some of the key question addressed by this reviewer. The new results and data incorporated to the article, improve clearly its quality, support some of the key claims and weakness some of the conclusion that were speculative. This is clearly an improved version of the article.